# Modeling glacial and fluvial landform evolution at large scales using a stream-power approach

Stefan Hergarten[1]

[1]Institut für Geo- und Umweltnaturwissenschaften, Albert-Ludwigs-Universität Freiburg, Albertstr. 23B, 79104 Freiburg, Germany

**Correspondence:** Stefan Hergarten
(stefan.hergarten@geologie.uni-freiburg.de)

**Abstract.** Modeling glacial landform evolution is more challenging than modeling fluvial landform evolution. While several numerical models of large-scale fluvial erosion are available, there are only a few models of glacial erosion, and their application over long time spans requires a high numerical effort. In this paper, a simple formulation of glacial erosion which is similar to the fluvial stream-power model is presented. The model reproduces the occurrence of overdeepenings, hanging valleys, and steps at confluences at least qualitatively. Beyond this, it allows for a seamless coupling to fluvial erosion and sediment transport. The recently published direct numerical scheme for fluvial erosion and sediment transport can be applied to the entire domain, where the numerical effort is only moderately higher than for a purely fluvial system. Simulations over several million years on lattices of several million nodes can be performed on standard PCs. An open-source implementation is freely available as a part of the landform evolution model OpenLEM.

## 1 Introduction

Glaciers have played a major part in shaping several orogens on Earth. In contrast to fluvial erosion, however, glacial erosion has not been extensively considered in modeling large-scale landform evolution. The greater prevalence of rivers and their relevance for life and society almost everywhere on Earth may be the main reason for this imbalance. River dynamics have been studied quantitatively for more than a century not only scientifically, but also in numerous engineering projects. As pointed out by Harbor (1989), glacier dynamics have also been studied quantitatively since the end of the 19th century, but received little attention. Nowadays, glacier dynamics are still more difficult to observe than fluvial dynamics, and there are larger uncertainties about the relevance of the involved processes on glacial landform evolution (e.g., Alley et al., 2019). As a consequence, the respective models are not only mathematically and numerically more complex than models of fluvial landform evolution, but also less well constrained.

Models of large-scale fluvial landform evolution (for an overview, see, e.g., Willgoose, 2005; Wobus et al., 2006) are typically based on a simple expression for the erosion rate which dates back to studies of longitudinal river profiles by Hack (1957) and was introduced in numerical landform evolution modeling by Howard (1994). This relation is often referred to as the stream-power law or as the stream-power incision model and considers the erosion rate $E$ as a function of the upstream

catchment size $A$ and the channel slope $S$ in the form

$$E = KA^m S^n. \tag{1}$$

All dependencies of $E$ except for $A$ and $S$ are subsumed in the lumped parameter $K$, called erodibility. This also includes the geometry of the river's cross section, so that rivers can be considered as linear elements. The values of the exponents $m$ and $n$ are believed to be more or less universal. Equation (1) is occasionally written in the form

$$E = K \left( A^\theta S \right)^n \tag{2}$$

with the concavity index $\theta = \frac{m}{n}$. In contrast to the absolute values of $m$ and $n$, $\theta$ can be determined from the shape of longitudinal river profiles if the erosion rate is uniform along the river. Thus, the value of $\theta$ is well-constrained. Most modeling studies either use the value $\theta = 0.5$ originally found by Hack (1957) or a slightly lower reference value $\theta = 0.45$ (e.g., Whipple et al., 2013; Lague, 2014).

While the stream-power incision model is limited to scenarios where all material that is detached from the river bed is
immediately excavated, some extensions towards sediment transport were developed. Recently, a generic formulation

$$\frac{E}{K_d} + \frac{Q}{K_t A} = A^m S^n, \tag{3}$$

called shared stream-power model, was presented (Hergarten, 2020), where $Q$ is the sediment flux (volume per time). This model contains two parameters beyond the exponents $m$ and $n$, where $K_d$ describes the erodibility in absence of transported sediment ($Q = 0$) and $K_t$ the ability to transport sediment at zero erosion rate ($E = 0$). Since the physical units and the
interpretation in the context of Hack's findings of $K_d$ and $K_t$ are identical, both can be considered as erodibilities, although $K_t$ is rather a transport coefficient.

Both the stream-power incision model and the shared stream-power model (as well as the mathematically equivalent model of Davy and Lague, 2009) can be implemented very efficiently in landform evolution models using fully implicit schemes (Hergarten and Neugebauer, 2001; Braun and Willett, 2013; Yuan et al., 2019; Hergarten, 2020). Here the maximum time
increment is not limited by the numerical stability, but only by changes in the flow pattern. Beyond this, the numerical effort per time step increases only linearly with the size of the grid for the schemes mentioned above, which allows for simulations on large grids over long time spans.

A comparable representation of glacial erosion where the erosion rate can be directly computed from properties of the topography is not yet available. Contemporary models involve at least the thickness of the ice layer and its velocity at each
point of the topography. The shallow-ice approximation (Fowler and Larson, 1978; Hutter, 1980; Cuffey and Paterson, 2010) is widely used in this context. The flow follows the direction of the steepest descent of the ice surface, and the depth-averaged horizontal velocity is decomposed into a sliding velocity $v_s$ and a depth-averaged deformation velocity $v_d$. The latter can be obtained directly by combining Glen's flow law with the shallow-ice approximation, which yields a power-law dependence on on the thickness $h$ of the ice layer and on the slope $S$ of the ice surface,

$$v_d \sim h^{\psi+1} S^\psi, \tag{4}$$

where $\psi$ is the exponent of Glen's flow law (typically $\psi \approx 3$, e.g., Cuffey and Paterson, 2010). In contrast, developing a similar relation for the sliding velocity $v_s$ relies on further assumptions. The simple relation proposed by Budd et al. (1979) involves the shear stress and the effective normal stress at the bed, which is in general lower than the static pressure of the ice column due to the pressure of the meltwater. The simplest models assume that the water pressure is proportional to the static pressure of the ice column, i.e, to the thickness $h$ (e.g., Harbor et al., 1988; Braun et al., 1999; Deal and Prasicek, 2021). Then the sliding velocity follows a relation that is very similar to Eq. (4),

$$v_s \sim h^{\psi-1} S^{\psi} \tag{5}$$

(e.g., Deal and Prasicek, 2021).

The erosion rate is usually assumed to depend on the sliding velocity also by a power-law relation

$$E \sim v_s^l, \tag{6}$$

where the linear version ($l = 1$) originally suggested by Hallet (1979) is still widely used. It should be noted that this relation involves a large uncertainty not only in itself, but also since it inherits the uncertainty from the sliding velocity.

The product of ice thickness and total velocity, $h(v_s + v_d)$, defines the ice flux per unit width. Inserting this property into the mass balance of the glacier yields a differential equation for the thickness $h$. If the topography of the ice surface ($H + h$, where $H$ is the topography of the bedrock) is considered as the variable, it is a diffusion equation with a high and strongly variable diffusivity. This property makes its numerical treatment challenging and allows only for small time increments.

While some fundamental studies focusing on modeling the evolution of longitudinal and transversal valley profiles (Oerlemans, 1984; Harbor et al., 1988; Harbor, 1992) even date back to the 1980s, the model ICE-CASCADE provided the first implementation of this concept in a large-scale landform evolution model (Braun et al., 1999). While this model has been applied in several studies, the iSOSIA (integrated Second-Order Shallow Ice Approximation) model introduced by Egholm et al. (2011) is a step towards a more realistic description of ice flow without solving the full three-dimensional equations of flow. As a major point, it also takes into account longitudinal and transverse stresses, which impedes strong local gradients in ice velocity and the incision into the bedrock along thin lines. This problem had to be fixed in ICE-CASCADE by a more heuristic approach based on the curvature of the topography. Both models were extended by models of meltwater dynamics (Herman et al., 2011; Egholm et al., 2012), which helps to constrain the sliding velocity as discussed above. However, both models still require small time increments and are computationally expensive.

## 2 A stream-power law for glacial erosion

The model presented in this section is in its spirit very similar to the sliding ice incision model (SIIM) proposed recently by Deal and Prasicek (2021). The main difference is that Deal and Prasicek (2021) developed an approximation for the total velocity $v_s + v_d$ over the typical range of ice thickness of alpine glaciers that can be treated analytically, while the limit of zero thickness is the starting point here. Although this approach is clearly less accurate, it will turn out to be useful when combined with fluvial erosion. The extension towards finite ice thickness will be the subject of Sect. 6.

According to Eqs. (4) and (5), the ratio of $v_d$ and $v_s$ is proportional to $h^2$. Thus, the relative contribution of deformation to the total flux converges to zero in the limit of a thin layer, provided that the ice is not frozen to the bedrock. So let us assume for the moment that the entire ice flux is dominated by sliding. If we consider a rectangular cross section of a width $w$, the total ice flux (volume per time, not per unit width) is

$$q_i = w h v_s. \tag{7}$$

The thickness $h$ can be eliminated by combining Eqs. (5) and (7), which yields

$$v_s \sim \left( \frac{q_i}{w} \right)^{\frac{\psi - 1}{\psi}} S. \tag{8}$$

Then the erosion rate (Eq. 6) is

$$E \sim \left( \left( \frac{q_i}{w} \right)^{\frac{\psi - 1}{\psi}} S \right)^l. \tag{9}$$

In order to obtain an expression similar to the fluvial stream-power law (Eq. 1 or 2), we need an estimate of the width $w$. In the 1D model of Prasicek et al. (2020) with a dendritic glacier network, the width was expressed in terms of the upstream flow length. This approach was justified by results of Bahr (1997) who found a power-law relationship

$$w \sim L^\xi \tag{10}$$

with $\xi = 0.6$, where $w$ is the mean width and $L$ the total length of the glacier. Prasicek et al. (2020) obtained the same relationship with a slightly lower exponent of $\xi = 0.58$ from an analysis of 52,000 glacier polygons. Equation (10) cannot be transferred directly to the width at any point and the upstream flow length at this point. However, if we assume that it holds for each point in terms of actual width and upstream flow length, it is also valid for the mean width and the total flow length. So the proposition of Prasicek et al. (2020) that the scaling within individual glaciers also follows Eq. (10) is at least not at odds with the observations of Bahr (1997), though it cannot be concluded solely on these. In absence of any model that is better supported by data, Eq. (10) is used.

Prasicek et al. (2020) also found that glaciers follow the fundamental relationship

$$L \sim A^\eta \tag{11}$$

originally proposed by Hack (1957) for rivers, although with a slightly lower exponent $\eta$. While Hack (1957) found $\eta = 0.6$ and a later, more comprehensive study of Rigon et al. (1996) $\eta = 0.56$, Prasicek et al. (2020) obtained $\eta = 0.52$ from the drainage pattern of a previously glaciated region, suggesting that glacier patterns are more straight than rivers. Combining Eqs. (10) and (11) yields

$$w \sim A^\alpha, \tag{12}$$

where $\alpha = \eta \xi \approx 0.30$ with the values suggested by Prasicek et al. (2020).

If the rate of ice production was constant over the entire upstream catchment, $q_i$ would be proportional to $A$, and thus

$$w \sim q_i^\alpha. \tag{13}$$

As argued by Deal and Prasicek (2021), it is plausible to assume that variations in width along a glacier depend on the ice flux instead of the upstream catchment size. Here we should keep in mind that Eq. (12) cannot be derived directly from available

data, but is not in conflict with those. This property is not affected when replacing Eq. (12) by Eq. (13) if we assume that the rate of ice production follows a similar function along the glacier for all glaciers. So Eq. (13) does not rely on a constant rate of ice production, but the main reason for using it in the following is still the absence of any model that is better supported by data.

Using Eq. (13), the erosion rate (Eq. 9) turns into

125 $$E \sim \left( q_i^{\theta_g} S \right)^l \tag{14}$$

with

$$\theta_g = (1 - \alpha) \frac{\psi - 1}{\psi}. \tag{15}$$

Equation (14) has the same shape as Eq. (2), where $l$ takes the role of $n$. So Eq. (14) can be interpreted as a stream-power law for glacial erosion.

The glacial concavity index $\theta_g$ is about 0.47 for $\psi = 3$, which is strikingly close to the widely used values $\theta = 0.45$ or $\theta = 0.5$ for fluvial erosion. In contrast, Deal and Prasicek (2021) obtained a considerably lower glacial concavity index $\theta_g = \frac{1}{3}$ (for $\alpha = 0.25$, $\theta_g = 0.31$ for $\alpha = 0.3$). So the approach of Deal and Prasicek (2021) predicts a weaker increase of the erosion rate with increasing ice flux than the approach proposed here, owing to the different treatment of sliding and deformation. The approximation proposed by Deal and Prasicek (2021) takes into account an increasing ice flux results in a decreasing relative

contribution of sliding to the total velocity. This reduces the increase in erosion rate with ice flux. In contrast, the approach presented here is limited to the sliding-dominated regime and thus obviously weaker. In turn, $\theta_g$ is so close to typical values of $\theta$ for rivers that we can assume $\theta_g = \theta$. This considerably facilitates the formulation, the numerical implementation, and the interpretation of erodibilities. However, we have to think how to include the effect of deformation later (Sect. 6).

While Eq. (14) is already very similar to the fluvial model (Eq. 2), the main difference is that the latter is written in terms of

140 catchment size instead of flux for historical reasons. Since erosion rates rather depend on discharges than on catchment sizes, the erodibility $K$ is a lumped parameter that already includes precipitation implicitly. If we assume that a given erodibility $K$ refers to a uniform reference precipitation $p_0$, the long-term mean discharge from a catchment of size $A$ is $q = p_0 A$. Then Eq. (2) written in terms of $q$ instead of $A$ should also be applicable to non-uniform precipitation. However, we would either have to carry $p_0$ through all equations or to redefine the erodibility in the form $K p_0^{-m}$.

In order to avoid this inconvenience, a new terminology is introduced. Let us define

$$A_{eq} = \frac{q}{p_0} \tag{16}$$

and denote the result the catchment-size equivalent of the discharge. It describes the catchment size that is needed to generate the discharge $q$ at the uniform reference precipitation $p_0$. While it may not be intuitive to measure discharge in terms of area, the advantage is that all relations for erosion remain valid for non-uniform precipitation. For a uniform precipitation $p = p_0$, $A_{eq} = A$, while $A_{eq}$ differs from the geometric catchment size $A$ else. Since all subsequent relations (except for the $\chi$ transform, Eq. 29) are based on discharges instead of catchment sizes, the subscript is omitted in the following, so the symbol $A$ refers to catchment-size equivalents.

The ice flux $q_i$ can also be measured as a catchment-size equivalent according to

$$A_i = \frac{q_i}{p_0}. \tag{17}$$

In principle, any reference precipitation $p_0$ can be used here. In a coupled model of glacial and fluvial erosion, however, $p_0$ must be the same for both components since the catchment-size equivalent is nothing but an alternative physical unit for measuring fluxes. Using the catchment-size equivalent of the ice flux, Eq. (14) can be written in the same form as Eq. (2),

$$E = K_g \left( A_i^\theta S \right)^l. \tag{18}$$

As mentioned above, both exponents $l$ and $n$ are not well-constrained. However, values of 1 or 2 are often used in both cases. So it is convenient to assume $l = n$. The glacial stream-power law can then be written in the same form as Eq. (1),

$$E = K_g A_i^m S^n. \tag{19}$$

where the exponents $m$ and $n$ are the same as for fluvial erosion. Then the erodibilities $K_g$ and $K$ have the same meaning and the same physical unit. As an example, $K_g = K$ would mean that a glacier at a given ice flux and a given slope of the ice surface has the same erosion rate as a river at the same discharge and the same channel slope.

## 3 Implementation in a landform evolution model

While the glacial stream-power model developed in the previous section is similar to the fluvial model, two specific aspects have to be taken into account in a numerical implementation in a landform evolution model. First, the slope $S$ refers to the ice surface and not to the bedrock. This is in principle also true for the fluvial model (with the water level), but there the difference can be neglected at large scales. In turn, at least some characteristic features of glacial erosion such as overdeepenings require a difference between the slopes of the ice surface and the bedrock. Since introducing the ice surface as an additional variable would cost most of the model's simplicity, the implementation developed in this section still uses the approximation of small thickness from the previous section and assumes that both slopes are the same. The extension towards finite thickness will be discussed in Sect. 6.

Second, rivers and glaciers are considered as linear elements. While the width of individual rivers has a minor effect on fluvial landforms, U-shaped valleys as the perhaps most characteristic glacial feature cannot arise if erosion acts along a thin line. So the finite width of glaciers must be taken into account, although Eq. (19) refers to a line. Since Eq. (19) involves the

total ice flux through the entire cross section area and not the flux per unit width, a glacier should be described by a single cardinal flow line (like a river) instead of a parallel flow pattern, but its erosion must act over a wider domain. So let us start from a dendritic flow pattern on a discrete lattice as typically used in large-scale fluvial landform evolution models. Each cell of the grid has a unique flow direction towards the neighbor with the steepest descent, and the respective slope defines the channel slope $S$.

Let us assume steady-state conditions for the fluxes of water and ice, although this is, strictly speaking, not justified for the ice if the climate changes rapidly. The total flux $A$ (water and ice) and the ice flux $A_i$ (both measured in terms of catchment-size equivalent) are then given by the steady-state balance equations

$$A = s\frac{p}{p_0} + \sum_{\text{donors}} A, \tag{20}$$

$$A_i = s\frac{p_i}{p_0} + \sum_{\text{donors}} A_i, \tag{21}$$

where $p$ is the total precipitation, $p_i$ is the part of $p$ that is converted into ice, and $s$ is the pixel area of the considered grid cell. The sum extends over all neighbors that deliver their discharge to the considered site, called donors in the following.

This concept is widely used in modeling fluvial erosion. However, the spatial scales are different here. At least small rivers are usually narrower than the mesh width of the lattice. As this may cause problems for detachment-limited erosion in combination with hillslope processes, approaches taking into account that rivers cover only a part of the pixels of the grid were developed (Howard, 1994; Perron et al., 2008; Pelletier, 2010). Glaciers are, however, often wider than typical mesh widths of some tens to hundreds of meters. Thus, glacial erosion acts over an area around the cardinal flow path, called swath in the following. The scaling arguments developed in the previous section suggest that the width of the swath is a function of the ice flux according to Eq. (13). This concept is in principle the same as following the cardinal flow path with a pen of a variable width $w$.

Since Eq. (19) only describes the erosion rate at the cardinal flow path, an assumption must be made for the other points of the swath. At this stage, it is not possible to recover a physically justified representation of the across-valley variation in erosion from the simplifications already made. The goal must be to achieve that all points of the swath follow the erosion dictated by the cardinal flow line, taking into account that glacial valleys are typically U-shaped.

Transferring a suitable property directly from the cardinal flow line to the entire swath is the simplest idea. Erosion rate, surface height, and ice flux are candidates here. Figure 1 illustrates these versions, starting from a V-shaped valley. Assuming that all points across the swath share the same erosion rate comes closest to the spirit of the model, but is unable to change the shape of a valley. So V-shaped fluvial valleys would just incise more rapidly if $K_g > K$.

Extending the surface elevation to all points across the swath can be seen as the opposite limiting case, where V-shaped valleys would immediately turn into U-shaped valleys with a flat floor. This may be questionable with regard to estimates in the order of magnitude of 100,000 years for the conversion to a U-shaped valley (Harbor et al., 1988). More important here, this approach converts the dendritic flow pattern towards the cardinal flow line into a parallel flow pattern over long times. However, parallel flow does not contribute to the flux at the cardinal flow path and thus reduces the erosion rate artificially since the glacial stream-power law (Eq. 19) refers to the total ice flux and not to the flux per unit width. Moreover, serious problems would occur

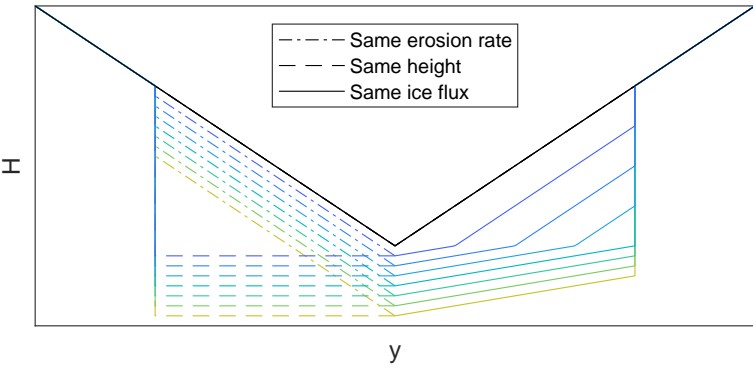

**Figure 1.** Different concepts of extending erosion from the cardinal flow path to a wider swath in order to transform a V-shaped valley (black line) into a U-shaped valley. Lower lines (from blue to yellow) describe increasing time. Not to scale.

if sediment transport is taken into account (Sect. 5), since a huge amount of material would be transported towards the cardinal flow line instantaneously.

The third version – transferring the ice flux $A_i$ to the swath and use this extended ice flux in the erosion law (Eq. 19) – works quite well. The key point is the the erosion rate at each point still depends on the local slope, so that the shape of the valley can adjust. Since the extended ice flux is typically much larger than the original flux, the valley floor becomes quite flat

through time. As illustrated in Fig. 1, however, the flanks of a V-shaped valley are not flattened continuously, but rather pushed outwards by a new, quite flat flat valley floor spreading from the cardinal flow path. This behavior is probably not completely realistic, but in turn, this concept is technically simple since it uses the original erosion model, where only the ice fluxes have to be modified.

The algorithm used for swath profiles across a given baseline based on the minimum distance (Hergarten et al., 2014) could

be used for drawing the swath and for assigning a reference point on the cardinal flow line to each point of the swath. In this study, however, a slightly different approach based on catchments is suggested.

Let $c_0$ be a point on the cardinal flow path, and $c_i$ for $i \geq 1$ the upstream points on the cardinal flow path, defined by the condition that $c_{i+1}$ is the biggest donor of $c_i$. Let $u$ be any donor of $c_0$, which is not on the cardinal flow path ($u \neq c_1$). The point $u$ is added to the swath and assigned to $c_0$ if the distance between $u$ and $c_i$ is not greater than $\frac{w_i}{2}$ for any $i \geq 0$, where

$w_i$ is the width according to Eq. (13) applied to the ice flux of the point $c_i$. After selecting the donors of $c_0$ that satisfy this condition, the same procedure is applied to all donors of these points. The procedure continues until no more donors that satisfy the condition are found.

However, there is no straightforward definition of cardinal flow lines on an absolute scale since dendritic flow patterns may cover a wide range of scales. Therefore, the procedure described above is applied to all points $c_0$ with a nonzero ice flux.

Finally, the largest value of $A_i$ (resulting from different starting points $c_0$) is taken for each point.

An example of the obtained swaths and the respective fluxes is shown in Fig. 2. The topography used here is a fluvial equilibrium topography with $K = 1$, $m = 0.5$, and $n = 1$ under uniform uplift $U = 1$ on a grid of $5000 \times 5000$ nodes. This

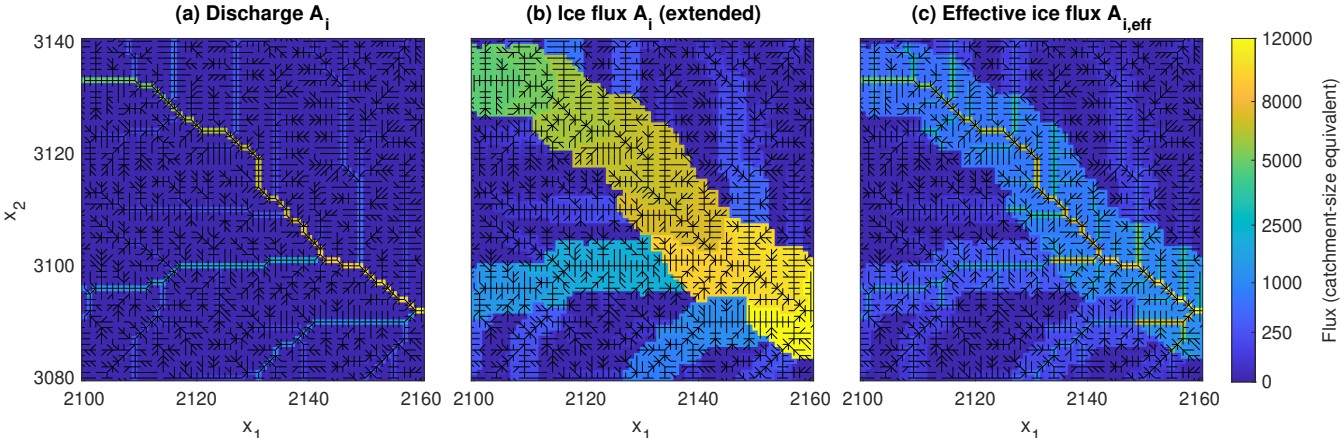

**Figure 2.** Illustration of the fluxes in the swath around the cardinal flow path. The domain corresponds to the small, white square in Fig. 3.

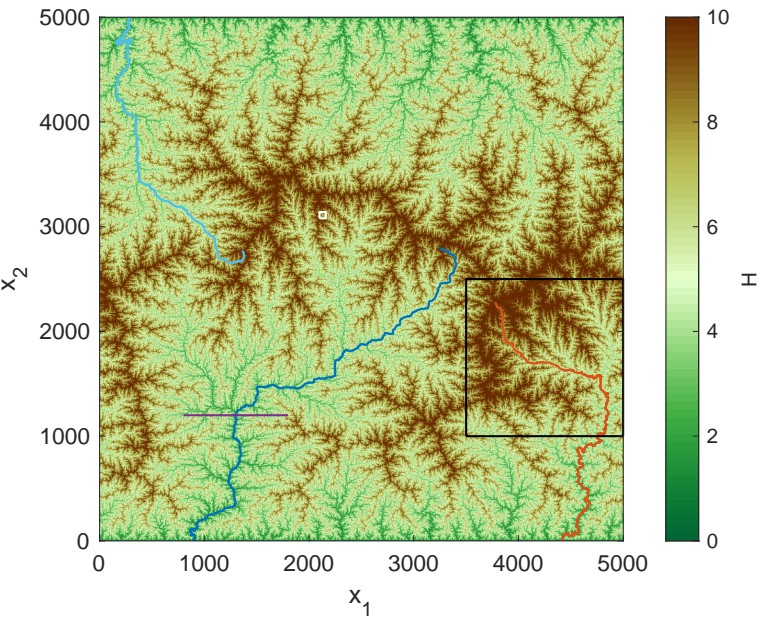

**Figure 3.** Fluvial equilibrium topography under uniform uplift (Hergarten, 2020, Fig. 1a). The small, white square depicts the domain of Fig. 2, and the black square the domain of Figs. 10 and 12. The blue and red lines show the three biggest rivers. The violet line refers to the profile shown in Fig. 5.

topography is shown in Fig. 3. It was already used by Hergarten (2020, Fig. 1a), where the only difference is a shift of the periodic eastern and western boundaries in such a way that the three biggest rivers do not cross the boundaries.

The parameter values defined above refer to nondimensional coordinates. The choice $m = 0.5$ and $n = 1$ is very convenient in this context since the horizontal length scale, the vertical length scale, and the time scale are independent then. The erodibility

has a unit of inverse time and directly defines the time scale. If we, e.g., assume a fluvial erodibility of $K = 2.5 \, \text{Myr}^{-1}$ (Robl et al., 2017), a unit of nondimensional time corresponds to 400,000 yr. The vertical length scale can be defined arbitrarily. If we define one vertical unit as 250 m, the maximum nondimensional surface height of about 16 corresponds to 4000 m. The uplift rate $U = 1$ (one vertical unit per time unit) is equivalent to $0.625 \, \text{mm} \, \text{Myr}^{-1}$ then. The horizontal length scale can even be chosen arbitrarily without any interference with the parameter values.

Numerical experiments revealed that the approach described above is still not able to suppress parallel flow completely. If the ice flux is extended to the swath, the slope of the quite flat valley floor finally approaches the channel slope of the cardinal flow line. While is this not relevant if only a cross section is considered as in Fig. 1, the channel slope will prevail at least at some locations in the swath in general. In order to support the dendritic flow pattern towards the cardinal flow line sufficiently, the across-valley slope should be greater than the channel slope of the cardinal flow line. This can be achieved by reducing the extended ice flux. As a simple approach, an effective ice flux obtained from a weighted geometric mean value

$$A_{\text{i,eff}} = A^\epsilon \, A_{\text{i}}^{1-\epsilon} \tag{22}$$

can be used in the glacial stream-power law, where $A_{\text{i}}$ is the extended ice flux and $A$ the (non-extended) total flux. The parameter $\epsilon$ defines the weighting of the two fluxes. Larger values of $\epsilon$ provide a better suppression of parallel flow patterns, but in turn make glacial valleys more V-shaped. In all numerical simulations performed during the preparation of this paper, $\epsilon = 0.25$ turned out to be a safe choice. Since $A$ typically decreases with increasing distance from the central flow line, this approach introduces some parabolic shape of the valley flow, which is not unrealistic. However, it should be kept in mind that this concept just enforces the swath to follow the erosion dictated by the cardinal flow line with a not completely unrealistic valley shape, but is still far off from predicting the evolution of valley shapes on a theoretically solid basis.

As a first example, Fig. 4 shows a steady-state glacial topography for $K_{\text{g}} = 1$. It was assumed that the entire precipitation is converted into ice ($p_{\text{i}} = p = p_0$). The factor of proportionality in Eq. (13) was set to unity (in units of the grid spacing), so $w = A_{\text{i}}^{0.3}$. The term steady state is seen in a loose sense here. After simulating a time span of 25 time units starting from the fluvial equilibrium topography, both the maximum and the mean surface height show no systematic trend any more, although they still oscillate due to local reorganization of the drainage pattern. Obtaining a steady state in the strict sense would, if possible at all, require a much longer time period, but this is not relevant for this study.

As expected, a strong downstream increase in valley width is the main difference towards the fluvial equilibrium topography. Due to the U-shape of the big valleys, the parts of the tributaries that are captured by the swath are almost flat, which is equivalent to lowering their base level compared to the fluvial topography. As a consequence, the surface heights of high regions decrease, although the same erodibility as for fluvial erosion was assumed.

Both the occurrence of wide, U-shaped valleys and the reduced elevation are also visible in the topographic profiles shown in Fig. 5. Beyond these, the slight V-shape of the valley floors arising from Eq. (22) is also visible. If we assume a typical grid spacing of 50 m, the largest valley is about 6 km wide. The height difference across the largest valley is about 0.08, corresponding to about 20 m if one vertical length unit is equivalent to 250 m (see Sect. 3). So the across-valley slope of

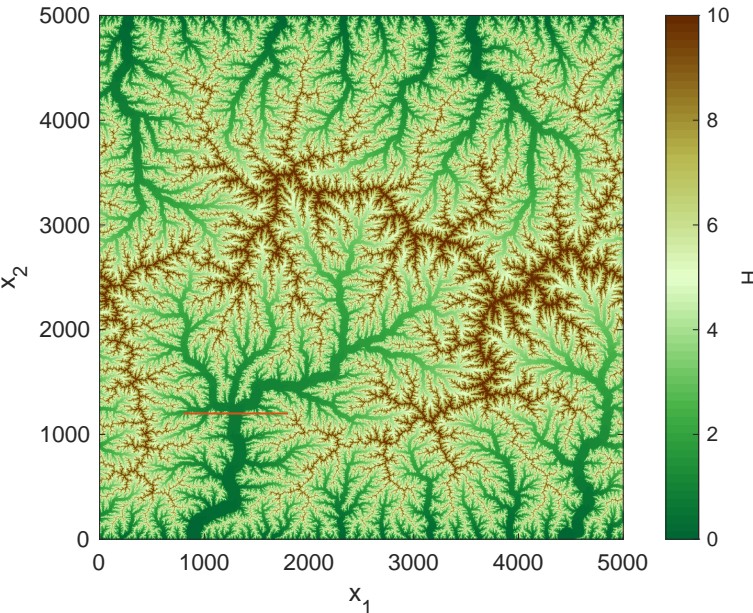

**Figure 4.** Glacial equilibrium topography under uniform uplift with the entire precipitation converted into ice. The red line refers to the profile shown in Fig. 4.

this valley is about 0.3 %. Smaller valleys, however, have a stronger residual V-shape. The valleys around $x_1 = 1000$ and $x_1 = 1500$, e.g., have across-valley slopes between 2 % and 6 %.

## 4 Fluvio-glacial systems

The ratio of $p_i$ and $p$ typically decreases with decreasing altitude. The equilibrium line altitude (ELA) defines the height where $p_i = 0$. Below the ELA, melting dominates, so $p_i$ is negative. Somewhere below the ELA, melting even compensates the contribution of tributaries, resulting in a downstream decrease in $A_i$.

A simple, linear model for the part of the total precipitation that is converted into ice,

$$p_i = p \min\left(\frac{H - H_e}{H_f - H_e}, 1\right), \tag{23}$$

is used in this study, where $H_e$ is the ELA and $H_f$ is the altitude where the entire precipitation is converted into ice. The surface height $H$ cannot be included easily in the fully implicit scheme for erosion, so it is evaluated at the beginning of the time step. This will, however, not strongly affect the stability of the implicit scheme and is a minor restriction of the maximum time increment compared to treating the flow directions in an explicit manner (Hergarten, 2020).

Figure 6 shows an example of the total flux and the ice flux of the three largest rivers from Fig. 3 for $H_e = 8$ and $H_f = 10$, corresponding to 2000 m and 2500 m, respectively, using the vertical scale defined above. Practically, $A_i$ is still not much lower than $A$ at the ELA. The point of maximum ice flux is found below the ELA due to the ice flux of tributaries. At this point,

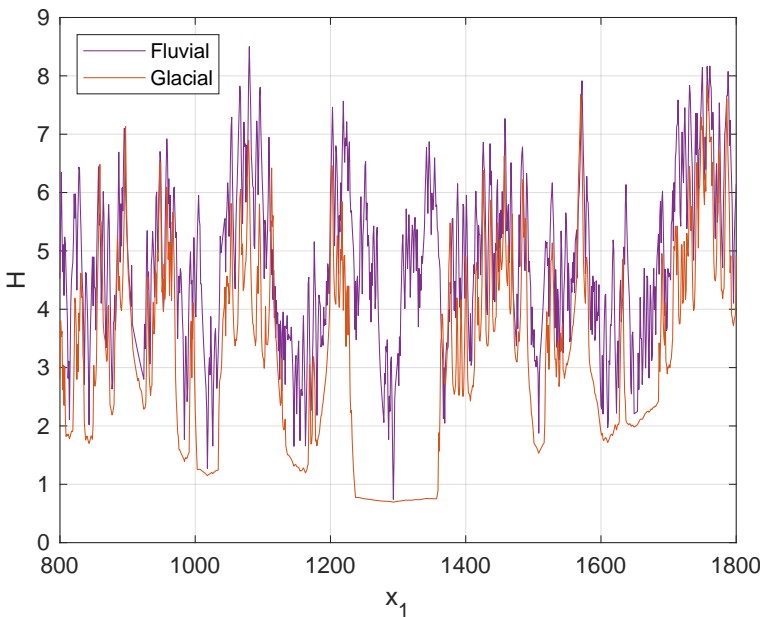

**Figure 5.** Topographic profiles along the lines depicted in Figs. 3 and 4.

$A$ is already more then three times greater than $A_i$ for the considered flow paths. This means that the discharge of water is considerably higher than the ice flux for a major part of the flow path.

Assuming that all sites with $A_i > 0$ are eroded glacially, while fluvial erosion only affects ice-free sites, is the simplest concept of a coupled model. It was used in the context of steady-state topographies by Prasicek et al. (2020). Figure 7 shows a topography obtained for $H_e = 8$ and $H_f = 10$, where the glacial erodibility, $K_g = 2$, is twice the fluvial erodibility ($K = 1$). This choice has no specific relevance here. The ratio of 2 is just convenient for recognizing steady-state longitudinal profiles visually. While the initial state (fluvial equilibrium) and the simulated time span ($t = 25$) are the same as in the example with full glaciation (Fig. 4), the topography is still far off from a steady state here.

Profiles along the three largest flow paths marked in Fig. 7 are depicted in Fig. 8 (lower solid lines). All profiles show a quite steep increase at the glacier terminus, which may even be a sharp front. It occurs because the ice flux approaches zero close to the glacier terminus. In this situation, a nonzero erosion rate can only be achieved if the decreasing ice flux is compensated by an increasing slope. This effect is amplified by the zero-thickness approximation that is still used at this stage, where the slope $S$ is the same for the bedrock and for the ice surface. However, it is neither the result of this approximation alone nor an inherent property of the stream-power approach since it also occurred the steady-state topographies obtained by Prasicek et al. (2020, Figs. 5 and 7) using the shallow-ice approximation with finite thickness.

Regardless of the question whether this behavior is realistic, including subglacial fluvial erosion would change this behavior. So offering an option for taking into account subglacial fluvial processes is not a disadvantage. Sediment transport by meltwater was included in the iSOSIA model quite soon (Egholm et al., 2012). While Beaud et al. (2016) developed a more elaborate

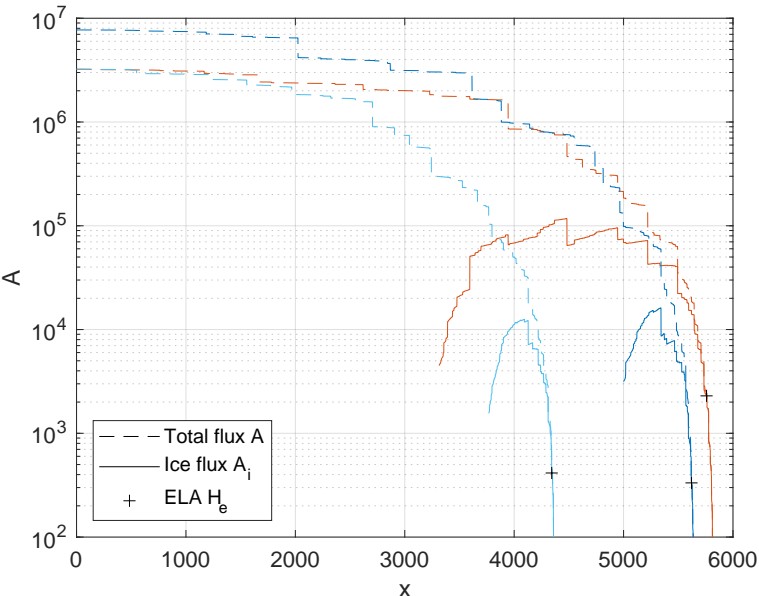

**Figure 6.** Total flux $A$ and ice flux $A_i$ of the flow paths shown in Fig. 3 for $H_e = 8$ and $H_f = 10$, both expressed in terms of their catchment-size equivalent. The along-stream coordinate $x$ starts from the boundary of the domain.

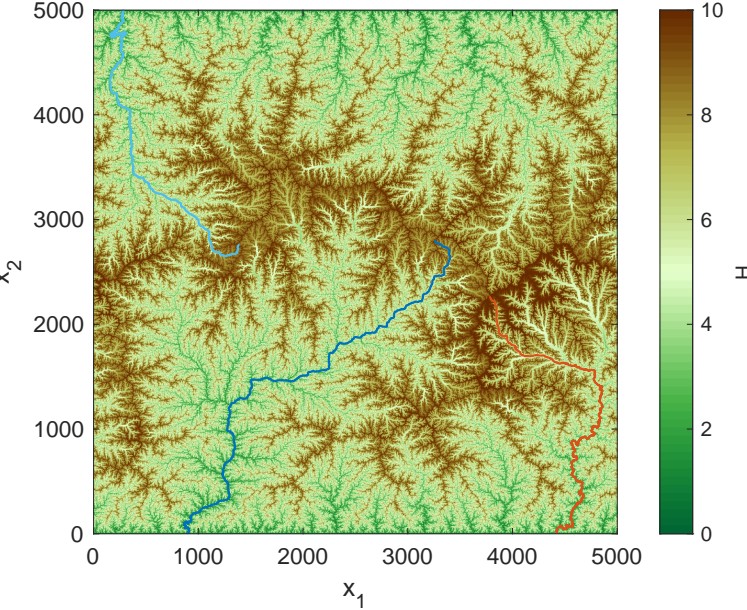

**Figure 7.** Topography at $t = 25$ for $H_e = 8$, $H_f = 10$, $K_g = 2$, and $K = 1$.

model for the incision by meltwater within narrow channels, erosion by subglacial fluvial processes is still one of the most challenging and controversial topics in the field of glacial erosion (e.g., Alley et al., 2019).

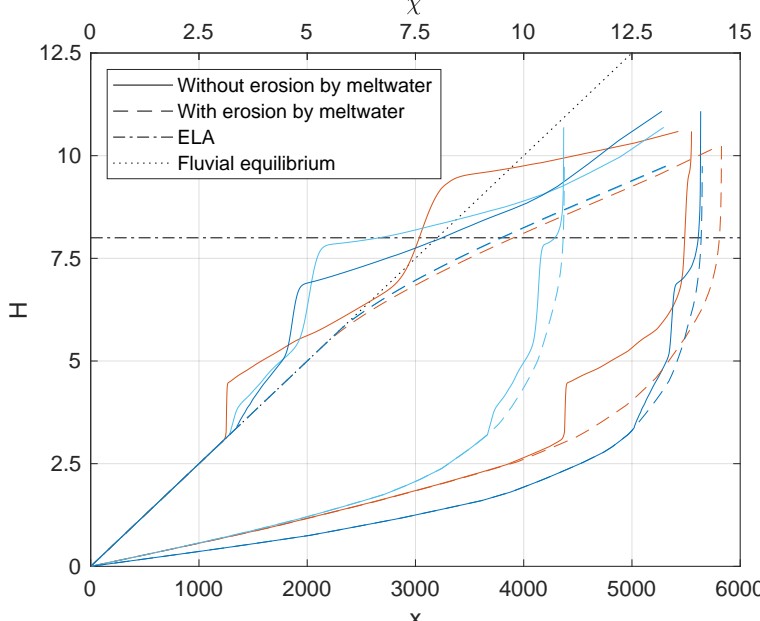

**Figure 8.** Longitudinal profiles of the largest flow paths in the fluvio-glacial topographies shown in Fig. 7 (solid lines) and Fig. 9 (dashed lines). The lower set of lines refers to the original along-stream coordinate $x$, while the upper set of lines depicts the $\chi$-transformed profiles. Both start from the boundary of the domain.

As a simple approach, it is proposed here to describe subglacial fluvial processes by the same equations as fluvial processes in rivers. This approach is clearly limited by not taking into account the water pressure explicitly. In fact, the apparent absence of any better model that can compete with the simplicity of the stream-power model is the only justification of this approach. The respective form of Eq. (1) reads

$$E = K_{\mathrm{f}} \left( A - A_{\mathrm{i}} \right)^{m} S^{n}, \tag{24}$$

where $A - A_{\mathrm{i}}$ is the the catchment-size equivalent of the meltwater flux. The respective erodibility $K_{\mathrm{f}}$ cannot be constrained easily since it obviously depends the topology and the cross-sectional geometry of the meltwater channels and on which fraction of the total meltwater flux reaches the valley floor. As discussed, e.g., by Egholm et al. (2012), this fraction may also depend on the thickness of the ice layer. While the flux $A - A_{\mathrm{i}}$ could be multiplied by the respective factor, it would also be possible to include it in the erodibility as a lumped parameter.

Adding the glacial erosion rate (Eq. 19) and the subglacial fluvial erosion rate (Eq. 24) yields

$$E = K_{\mathrm{g}} A_{\mathrm{i}}^{m} S^{n} + K_{\mathrm{f}} \left( A - A_{\mathrm{i}} \right)^{m} S^{n}. \tag{25}$$

This expression can be written conveniently in the form of Eq. (1) or Eq. (19),

$$E = K_{\mathrm{eff}} A^{m} S^{n}, \tag{26}$$

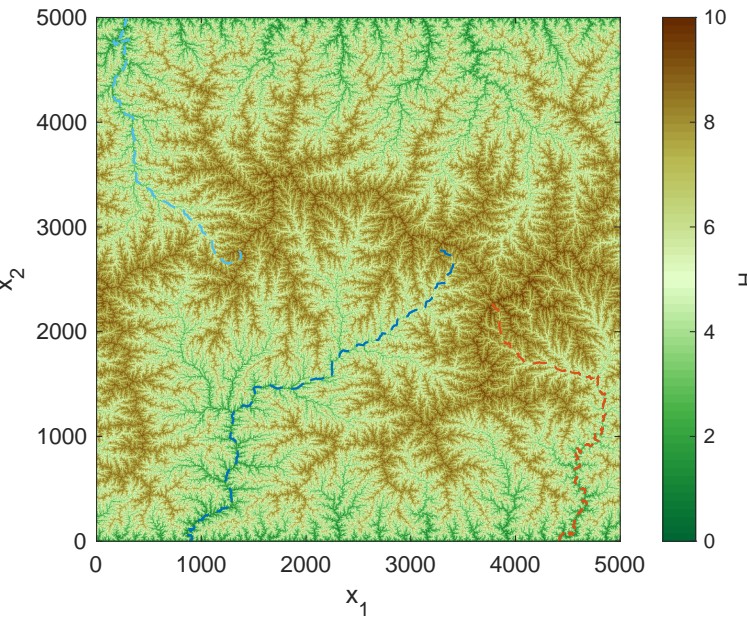

**Figure 9.** Topography obtained under the same conditions as in Fig. 7, but assuming that erosion by meltwater follows the same relation as fluvial erosion with the same erodibility.

with the effective erodibility

$$K_{\text{eff}} = \gamma^m K_{\text{g}} + (1 - \gamma)^m K_{\text{f}}, \tag{27}$$

where

$$\gamma = \frac{A_{\text{i}}}{A} \tag{28}$$

is the relative contribution of ice to the total flux.

In the following examples, the same erodibility $K_{\text{f}}$ is assumed for rivers and for subglacial fluvial processes, so $K_{\text{f}} = K$. This should, however, not be seen as an attempt to provide a realistic estimate, which would go beyond the scope of this study. Figure 9 shows a fluvio-glacial equilibrium topography obtained under the same conditions as the previous example, but including subglacial fluvial erosion. While the topography is qualitatively similar to that shown in Fig. 7, the upper regions are lower here. This is recognized more clearly in the longitudinal profiles (Fig. 8).

The $\chi$ transform introduced by Perron and Royden (2013) provides a simple way to analyze longitudinal river profiles quantitatively. It transforms the upstream coordinate $x$ to a new coordinate

$$\chi = \int A(x)^{-\theta} dx. \tag{29}$$

The $\chi$ transform eliminates the inherent concavity of river profiles arising from the upstream decrease in catchment size. Equilibrium profiles under spatially uniform conditions turn into straight lines with a slope of $\left(\frac{U}{K}\right)^{\frac{1}{n}}$. In our example ($K = 1$, $U = 1$), it is even the diagonal line $H = \chi$.

The upper lines in Fig. 8 depict the $\chi$-transformed profiles. Assuming $K_f = K$ and $K_g > K$ guarantees $K_{eff} \geq K$ every-where. As a consequence, the respective profiles are never above the fluvial equilibrium line $H = \chi$, in contrast to those without subglacial fluvial erosion. The transition from the fluvial regime to the fully glaciated domain, characterized by a straight line with a slope of $\frac{1}{2}$ (according to $K_g = 2$), is smooth. This would not necessarily be the case for $K_f < K$, where the profile may be steeper than the fluvial equilibrium profile close to the glacier terminus.

## 340  5   Sediment transport

The shared stream-power model (Eq. 3) provides a simple formulation for the entire range between the detachment-limited and transport-limited end-members of fluvial erosion. In principle, it just describes how sediment transport reduces the ability to erode the bedrock in such a way that equilibrium river profiles remain consistent with the findings of Hack (1957). While this is a rather generic concept, the mathematically equivalent model of Davy and Lague (2009) explicitly refers to the settling of
particles.

     While the model ICE-CASCADE assumes that transported sediment has no effect on glacial erosion (so detachment-limited erosion), the more comprehensive model iSOSIA assumes that a thick layer of sediments protects the bed against erosion. Since glacial sediments are transported at the velocity of the ice in this model, a large sediment flux results in a thick layer and thus reduces the rate of erosion. Although this is an oversimplified view on the interaction between glacial erosion and sediment
transport in the iSOSIA model, it illustrates that it is not very far away from the shared stream-power model in its spirit. Contemporary models of bedrock incision or sediment transport by subglacial fluvial processes (Beaud et al., 2016; Delaney et al., 2019) combine models for the flow in channels with erosion laws based on shear stress. This is not fundamentally different from physically based models of erosion and sediment transport in rivers, apart from the more complex distribution of pressure that drives the flow.

Anyway, these considerations show that extending both the ice and the meltwater component of the model in such a way that sediment transport reduces the ability to erode the bedrock could be useful. Writing the shared stream-power model (Eq. 3) as the presumably simplest generic model in this context individually for the ice and the meltwater components yields

$$\frac{E_g}{K_{d,g}} + \frac{Q_g}{K_{t,g} A_i} \;=\; A_i^m S^n, \tag{30}$$

$$\frac{E_f}{K_{d,f}} + \frac{Q_f}{K_{t,f}(A - A_i)} \;=\; (A - A_i)^m S^n. \tag{31}$$

The additional indices g and f refer to the glacial and subglacial fluvial (meltwater) component, respectively. So there are four erodibilities now, referring to glacial and fluvial incision and transport, respectively.

     At this point, we should be aware that this step adds new capabilities to the model, but also moves it further away from well-constrained relations. However, assuming $K_{t,g} \to \infty$ or $K_{t,f} \to \infty$ recovers the detachment-limited version for the respective component. So we do not lose anything by proceeding to the shared stream-power model.

Since the flow of meltwater should be confined to narrow channels, the best approach would be to consider the two sediment fluxes $Q_g$ and $Q_f$ as separate variables without any mixing. The numerical scheme proposed by Hergarten (2020) could in

principle be extended accordingly. However, this would require additional theoretical and numerical effort. Beyond this, the two sediment fluxes merge anyway, e.g., if material is deposited by the glacier due to decreasing transport capacity when approaching the glacier terminus and eroded again by the meltwater stream. Therefore, a simpler approach using a single sediment flux is suggested. According to Eqs. (30) and (31), the total erosion rate $E = E_g + E_f$ follows the relation

$$E + \frac{K_{d,g} Q_g}{K_{t,g} \gamma A} + \frac{K_{d,f} Q_f}{K_{t,f}(1-\gamma)A} = K_{d,eff} A^m S^n, \tag{32}$$

where the definition of the effective erodibility for incision,

$$K_{d,eff} = \gamma^m K_{d,g} + (1-\gamma)^m K_{d,f}, \tag{33}$$

is the same as in Eq. (27). If the total sediment flux $Q = Q_g + Q_f$ is given, the fraction of the stream power spent for sediment transport depends on how $Q$ is distributed. Let us assume an optimized distribution in the sense that this fraction is minimized. It is easily recognized that this is the case if either the glacier or the meltwater carries the entire load, and thus

$$E + \min\left(\frac{K_{d,g}}{K_{t,g}\gamma}, \frac{K_{d,f}}{K_{t,f}(1-\gamma)}\right) \frac{Q}{A} = K_{d,eff} A^m S^n. \tag{34}$$

This equation can be written in the same form as the original shared stream-power model (Eq. 3),

$$\frac{E}{K_{d,eff}} + \frac{Q}{K_{t,eff}A} = A^m S^n, \tag{35}$$

with

$$K_{t,eff} = K_{d,eff} \max\left(\gamma \frac{K_{t,g}}{K_{d,g}}, (1-\gamma) \frac{K_{t,f}}{K_{d,f}}\right). \tag{36}$$

Figures 10 and 11 provide a numerical example. Since sediment transport is more interesting in transient states than in a steady state, a declining ELA was chosen here. The ELA starts from $H_e = 16$ (slightly higher than the maximum surface height of the fluvial equilibrium topography) and decreases at a rate of 4 (four times faster than the uplift), so that an ELA of $H_e = 8$ (and $H_f = 10$) is reached at $t = 2$.

The fluvial erodibilities are set to $K_{d,f} = 2.6$ and $K_{t,f} = 1.625$ for rivers and meltwater. Their ratio is $\frac{K_{d,f}}{K_{t,f}} = 1.6$. This ratio is the sediment deposition coefficient in the notation of Davy and Lague (2009), where a recent analysis of steady-state topographies suggested 1.6 as a realistic value for $n = 1$ (Guerit et al., 2019, data supplement). Furthermore, these values satisfy the relation

$$\frac{1}{K_{d,f}} + \frac{1}{K_{t,f}} = \frac{1}{K_f} \tag{37}$$

with $K_f = 1$, which ensures that the fluvial equilibrium topography computed for the detachment-limited model is also in equilibrium here (Hergarten, 2020). For glacial erosion, it is assumed that bedrock incision is not more efficient as fluvial bedrock incision ($K_{d,g} = K_{d,f} = 2.6$), while the glacier transports sediment 10 times more efficiently than water ($K_{t,g} = 10 K_{t,f} = 16.25$).

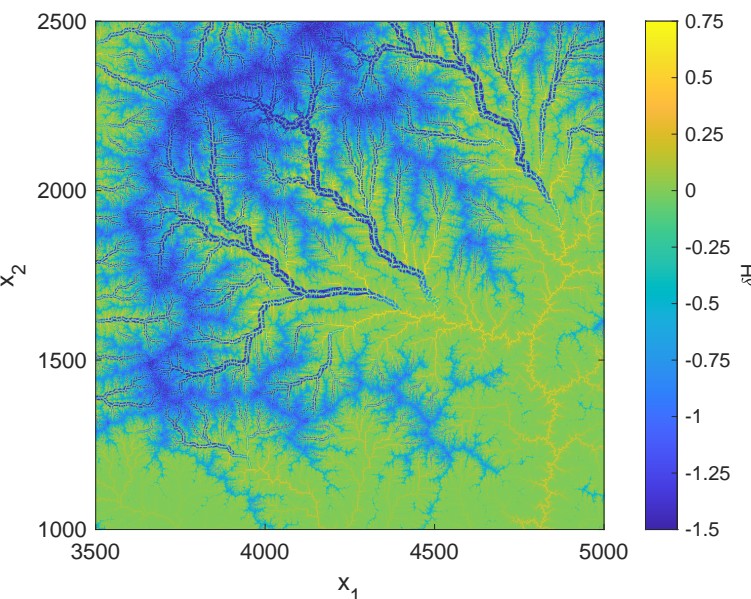

**Figure 10.** Change in surface height compared to the initial fluvial topography (black square in Fig. 3) obtained from the shared stream-power model at the time when the ELA reaches $H_e = 8$.

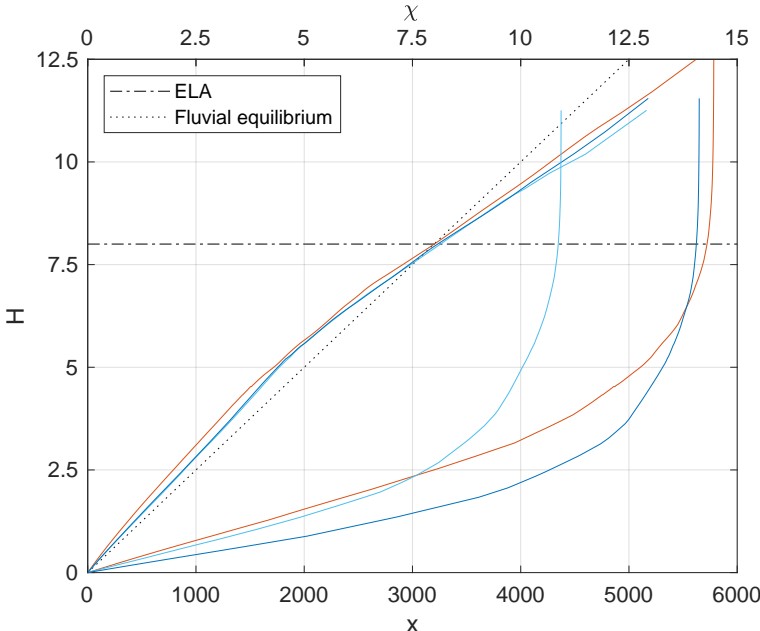

**Figure 11.** Longitudinal profiles of the largest flow paths, which are basically the same as in the fluvial topography in map view (Fig. 3). The lower set of lines refers to the original longitudinal coordinate $x$, while the upper set of lines depicts the $\chi$-transformed profiles.

Since the overall topography does not change fundamentally over the limited time span considered here, Fig. 10 shows the difference in height compared to the initial fluvial topography for a part of the domain. As expected, the strongest erosion occurs where V-shaped valleys turn into U-shaped valleys and at the fully glaciated high ranges. However, the most interesting effect is an increase in height along the original river, which concerns the U-shaped glacial valleys as well as rivers below the glaciers. In the glacial valleys, it arises from the rapid transformation towards a U-shape, which produces large amounts of sediment. In the non-glaciated part, it arises from the sediment flux from the glaciers, which is higher than that of the former rivers. It should, however, be noted than none of the changes in surface height describes a net deposition of sediments, but only reduced erosion. Since the total uplift is 2 ($U = 1$, $t = 2$), the maximum increase in elevation shown in Fig. 10 is less than half of the uplift, so that the minimum erosion is still more than half of the uplift.

These effects, which would not occur in the detachment-limited version, are also visible in the longitudinal profiles (Fig. 11). First, the fluvial parts of the profiles have become steeper compared to the initial fluvial topography due to the high sediment flux from the glaciated part. As the erosion rate in the glacial part is higher than in equilibrium, the glacier brings more sediment than the upstream part of the river brought under fluvial conditions. This increased sediment flux reduces the ability to erode the bedrock according to Eq. (3), and the fluvial erosion cannot follow the uplift any more.

The glaciated parts of the profiles are less steep than the initial fluvial profiles, but still steeper than it would be expected in equilibrium. Their slope in the $\chi$ plot is about 0.8, while Eq. (37) predicts an equilibrium slope of $\frac{1}{K_{d.g}} + \frac{1}{K_{t.g}} = 0.45$ with the glacial erodibilities assumed here. So the glaciated part is even steeper than the fluvial part if both are considered in relation to their equilibrium slope. The reason for the increased steepness is basically the same as for the fluvial part. Converting a V-shaped valley into a U-shaped valley goes along with a high erosion rate and thus yields a high amount of sediment. Although the efficiency of the glacier in transporting sediment was assumed to be ten times higher than for the rivers, the large amount of sediments limits the ability of the glacier to erode the bedrock during the conversion of the valley shape.

# 6   Finite ice thickness

The approach developed in Sect. 2 considers the limit of zero ice thickness. The finite thickness, however, has a strong influence on glacial landform evolution. Overdeepened valleys would be neither possible in the stream-power model proposed here nor in the original shallow-ice approximation in the limit of zero thickness. Beyond this, the ice surface defines a base level for the tributaries and may thus play an important part in the formation of hanging valleys.

A first estimate of the thickness $h$ can be obtained by combining Eqs. (7) and (8) in the form

$$q_i \sim w h \left( \frac{q_i}{w} \right)^{\frac{\psi - 1}{\psi}} S, \tag{38}$$

and thus

$$h \sim \left( \frac{q_i}{w} \right)^{\frac{1}{\psi}} S^{-1}. \tag{39}$$

In combination with Eq. (13), this yields

$$h \sim q_i^{\frac{1 - \alpha}{\psi}} S^{-1}. \tag{40}$$

Solving Eq. (40) numerically using an explicit scheme is not a big problem if the nodes are treated in upstream order. If the height of the bedrock surface, the ice flux, and the ice thickness of the downstream neighbor are known, the ice thickness of the respective node is obtained by solving a quadratic equation.

However, this concept still poses some challenges, in particular in combination with the consideration of the swath along the cardinal flow line. Therefore, a simple parameterization of $h$ by the ice flux alone without taking into account the slope is proposed here. While this concept is in its spirit consistent with the parameterization of the width by the ice flux alone, it should be emphasized that it leaves room for further development of better approaches.

For a straight profile ($S = $ const), the exponent in Eq. (40) is about 0.23 for $\alpha = 0.3$ and $\psi = 3$. On the other hand, the sliding
velocity is constant if the erosion rate is constant along the profile, and then Eqs. (7) and (13) immediately yield $h \sim q_\mathrm{i}^{1-\alpha}$. So the exponent is about 0.7 for a uniform erosion rate.

However, Eq. (40) still relies on the approximation of small thickness in the sense that the total ice flux is dominated by sliding. Adopting the approximation introduced by Deal and Prasicek (2021) would indeed be an advantage at this point since it captures sliding and deformation over the typical thickness range of alpine glaciers. However, the limit of small thickness
was preferred in Sect. 2 since it is consistent with using the same concavity index for fluvial and glacial processes, which considerably simplifies the formulation and the implementation. As discussed in Sect. 2, neglecting deformation increasingly underestimates the ice flux at large thickness. So the increase of thickness with ice flux is described well by Eq. (40) only for small thickness, while the increase becomes weaker for large thickness.

Taking this into account, a parameterization of the thickness by the ice flux alone should rather be in the lower range between
the two extremes 0.23 and 0.7 for the exponent found above. The simplest approach is assuming that the exponent is equal to $\alpha$ (here about 0.3), which means that glaciers have a constant thickness-to-width ratio. So let us assume $h \sim w$ in the following, although the model itself is still open for improved scaling relations.

Three effects of the finite thickness can be included in the model. First, the ice surface $H + h$ can be taken into account in the glacial mass balance (Eqs. 21 and 23), as it is usually done in glacial models. Since already $H$ has to be taken at the beginning
of the time step in Eq. (23), $h$ is treated the same way without affecting the stability of the scheme strongly.

The most important effect of a finite ice thickness is that the channel slope $S$ should refer to the ice surface $H + h$ instead of the bedrock surface $H$ as assumed in the previous examples. With the parameterization $h \sim w$, the implicit scheme for the erosion part can be maintained with the slope of the ice surface. Technically, this can be easily implemented by adding $h$ to $H$ before performing the erosion step and subtracting it afterwards. Here the scheme for drawing the swath around the cardinal
flow path described in Sect. 3 is particularly useful. It ensures that all points of the swath draining to a given point on the cardinal flow path, except for those located upstream on the cardinal path, have the same extended ice flux and thus also the same ice thickness. So the finite ice thickness has only a strong effect within the swath close to the glacier terminus and at confluences where the ice flux changes abruptly. In turn, it sets a higher base level for tributaries at the boundary of the swath, which facilitates the formation of steep walls and hanging valleys. However, local depressions in the ice surface may occur at
the boundary of the swath or along the cardinal flow path. These would cause a sediment flux opposite to the direction of the ice flux. In order to avoid this, local depressions in the ice surface should be filled.

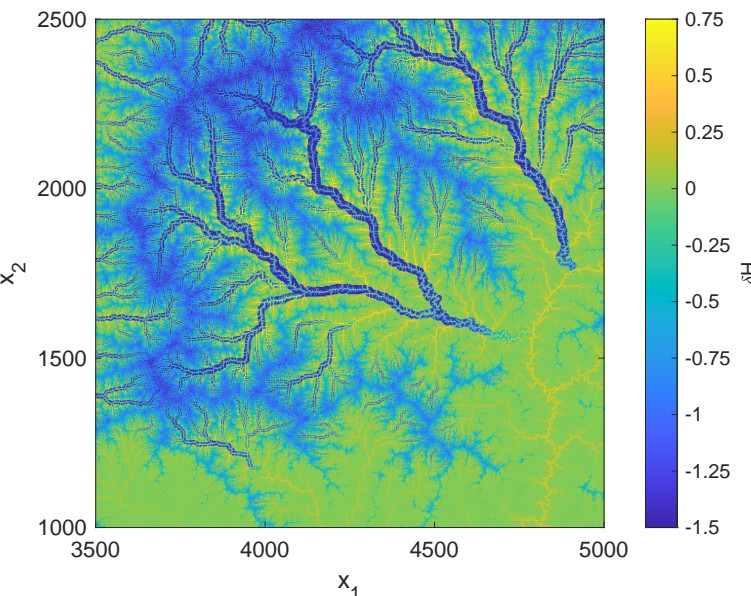

**Figure 12.** Change in surface height compared to the initial fluvial topography (black square in Fig. 3) obtained from the shared stream-power model with a thickness-to-width ratio of 0.05 at the time when the ELA reaches $H_e = 8$.

Figures 12 and 13 illustrate the effect of these two extensions on the example considered in Sect. 5. All parameters are the same, except that the thickness-to-width ratio is 0.05 in nondimensional coordinates. Using the previously suggested length scales of 50 m horizontally and 250 m vertically, the ratio would be 0.25 in reality.

While the slopes along the cardinal flow path are similar to the scenario with zero ice thickness, the glaciers become wider and advance further downstream here. This difference is related to taking into account the ice surface instead of the bedrock surface in Eq. (23), which defines the part of the precipitation that is converted into ice. This part has increased, which results in a higher ice flux and thus in an advancing of the glaciers.

The upper parts of the glaciated valleys feature smooth segments, which are interrupted by distinct steps. These steps occur
at major confluences. Due to the abrupt increase in ice flux at confluences, the ice thickness also increases abruptly. Over long times, however, erosion will smooth the ice surface, so that the step is transferred from the ice surface to the bedrock topography.

The overdeepening of the valley floor in the lower part of the glacier can be recognized well in the $\chi$-transformed profiles. The longitudinal shape of the bedrock surface looks quite irregular, with several short segments of strong overdeepening. This
localized overdeepening occurs if the glacier advances very slowly or the glacier terminus even stays at the same point for some time. The mechanism in the model is the same as for the confluences. Erosion attempts to flatten the steep front, and the gradient of the ice surface is transferred to the bedrock surface. In contrast to the steps at confluences, there is no persistent strong variation in ice flux here. So these local overdeepenings are transient structures which are slowly erased by erosion and filling when the glacier advances further.

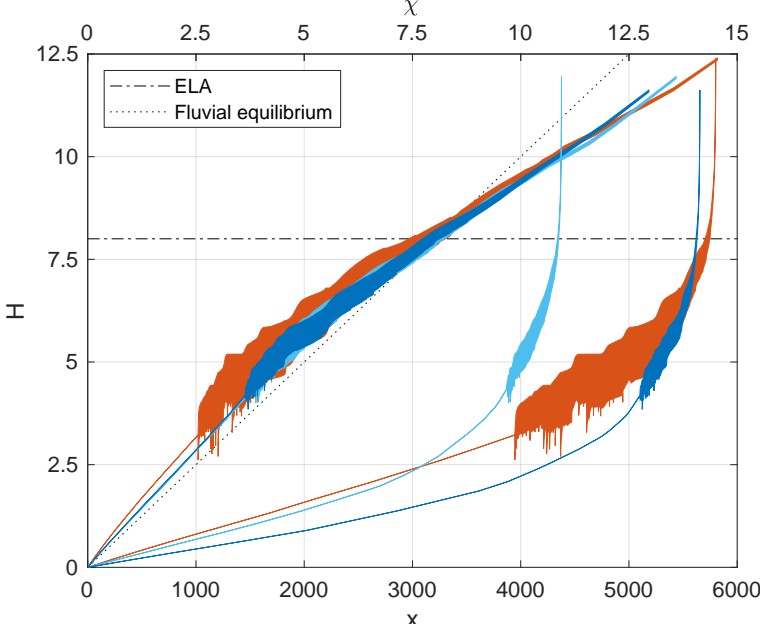

**Figure 13.** Longitudinal profiles of the largest flow paths in the topography shown in Fig. 12. The filled areas depict the ice layer. The lower set of lines refers to the original longitudinal coordinate $x$, while the upper set of lines depicts the $\chi$-transformed profiles.

The third point where the model could be improved by the finite ice thickness is the separation of the total flow velocity into a deformation velocity $v_d$ and a sliding velocity $v_s$. While the ice flux (Eq. 7) depends on the sum of both velocities, the erosion rate (Eq. 6) depends only on $v_s$. As discussed in the beginning of Sect. 2, the ratio of the two velocities is proportional to $h^2$. In the 1-D model of Prasicek et al. (2018), this relation was used for eliminating $v_d$ from the equations. Equation (7) then turns into

$$q_i = whv_s\left(1 + \beta h^2\right),\tag{41}$$

with a parameter $\beta$ depending on the rheology of the ice (for details, see Prasicek et al., 2018). The thickness $h$ cannot be eliminated easily then. If we, however, assume that the last term in Eq. (41) is only a small correction, we can insert the estimate of $h$ developed in this section there. Then all subsequent relations remain the same, but where the catchment-size equivalent $A_i$ of the ice flux is replaced by

$$\frac{A_i}{1 + \beta h^2} = \frac{A_i}{1 + \xi A_i^{2\alpha}}\tag{42}$$

with another parameter $\xi$. So the influence of the ice flux on erosion and sediment transport decreases if deformation becomes relevant. While Eq. (42) could easily be implemented, it introduces an additional parameter and will presumably not yield fundamentally different results. This extension is therefore not considered further in this study.

# 7 Numerical performance

The concepts developed in the previous sections were implemented in the open-source landform evolution model OpenLEM. A fully implicit scheme for the fluvial shared stream-power model was already available in OpenLEM before (Hergarten, 2020).

The behavior concerning the time increment $\delta t$ is basically the same as for the fluvial version. Practically, the accuracy is not limited by the accuracy of the implicit scheme itself, but by changes in the flow pattern, which are treated in an explicit way. While the flow directions are the only explicit component in the fluvial version, the glacial mass balance, i.e., the terms $H$ and $h$ in Eq. (23), is also treated in an explicit way in the fluvio-glacial version. This may introduce an additional limitation of the maximum time increment if the climatic conditions change rapidly.

The time-dependent simulations with the declining ELA and the finite ice thickness (Sect. 6) were performed with $\delta t = 10^{-2}$, $\delta t = 10^{-3}$, and $\delta t = 10^{-4}$. On a visual level (Figs. 12 and 13), the results were almost indistinguishable for $\delta t = 10^{-3}$ and $\delta t = 10^{-4}$, while a small difference was observed for $\delta t = 10^{-2}$. However, the difference is rather a small shift on the time axis than a principal difference in the shape of the glaciers and the resulting landforms. So $\delta t = 10^{-3}$ should be a safe choice, while larger values should also be possible if required, e.g., in long-term simulations. If we, e.g., assume a fluvial erodibility of $K = 2.5 \, \text{Myr}^{-1}$ (Robl et al., 2017), a unit of nondimensional time corresponds to 400,000 yr. So time increments of some hundred years appear to be safe, while some thousand years will also yield reasonable results.

However, the fluvio-glacial model requires a higher numerical effort per time step than the purely fluvial version. On the $5000 \times 5000$ grid, an increase in CPU time by a factor of about 1.7 was found for the fluvio-glacial shared stream-power model compared to the respective fluvial version even if the ice flux is zero everywhere. This factor increases to about 2.7 if the topography is completely glaciated. This factor, however, depends on the width of the glaciers. An increase of the factor of proportionality in Eq. (12) will result in an increasing numerical effort. In principle, the factors may also increase slightly for larger grids than the $5000 \times 5000$ nodes considered here since the width of the largest glaciers may also increase then. In turn, the implementation in OpenLEM used here is still in a preliminary state and leaves room for further optimization.

So the increase in numerical effort compared to the purely fluvial model is moderate, and simulations even over several million years are possible on standard PCs.

# 8 Strengths and weaknesses

The simplicity and the numerical efficiency are the most important strengths of the model proposed here. It allows for simulating combined fluvial and glacial landform evolution of entire mountain ranges over millions of years. This property also makes the model well-suited for considering multiple climatic and tectonic scenarios. While the recent study of Liebl et al. (2021) was limited by computing capacities, topographic signatures could be investigated in future for much larger domains with a higher spatial variability and over longer time spans where the effect of tectonics becomes relevant.

In turn, it must be admitted that the model is not a comprehensive model of glacial landform evolution. I was designed for valley glaciers from the beginning, and even the U-shape of typical glacial valleys was introduced explicitly. So the model cannot predict under which conditions U-shaped valleys occur or how flat the bottom of such a valley is. The model will even

fail completely, e.g., for piedmont glaciers. Here the glacier would keep its width after leaving the valley and would finally incise an overdeepened U-valley into the plain. Similarly, a plateau glacier would dissect the plateau into a network of U-shaped valleys.

The restriction to valley glaciers is closely related to the parameterization of the glacier width by the ice flux. This approximation decouples the valley shape from the longitudinal profile. Starting glaciation in a very steep fluvial topography, we should expect higher flow velocities and thus narrower glaciers than in a moderate topography. This effect is not captured by the parameterization of the glacier width, which also holds for the models of Prasicek et al. (2020) and Deal and Prasicek (2021). However, including the slope of the ice surface in the parameterization of the glacier width (and thickness) would cost

much of the model's simplicity.

    Accepting the parameterization of the glacier width, the stream-power law for bedrock incision by glaciers (Eq. 19) can in principle be derived from widely used relations. However, the subsequent steps rather leave solid ground. The idea of sharing the stream-power term $A_i^m S^n$ between bedrock incision and sediment transport is just adopted from the respective concept for fluvial erosion. Also, adopting the respective fluvial relations for incision and sediment transport by meltwater is only justified

by some similarity in the processes. So the relations going beyond glacial bedrock incision are rather based on intuition than on real-world data or on established relations.

    The parameters of the model are strongly lumped and do not refer directly to the physical properties of the ice and the bedrock. All $K$s occurring in the model have basically the same meaning and describe the ability of rivers, ice or meltwater to erode the bedrock or to transport sediment. So the model is not able to predict, e.g., under which conditions glacial erosion

is more efficient than fluvial erosion or how efficient subglacial fluvial processes are. These properties are prescribed by the parameter values. In turn, this rather phenomenological definition of the parameters may even by an advantage when investigating, e.g., how efficient glacial erosion must be compared to fluvial erosion in order to generate a buzzsaw-like erosion above the ELA, or how efficient subglacial fluvial erosion must be in order to have a considerable effect at large scales.

## 9   Conclusions

This study proposes a stream-power law for glacial erosion, which is in its spirit similar to the relation introduced recently by Deal and Prasicek (2021). While their model captures the contribution of deformation to the ice flux better, the approach proposed here has the advantage that the same concavity index may be assumed for fluvial and for glacial erosion. This property allows for a seamless combination with fluvial erosion, where fluvio-glacial erosion can also be included conveniently. Sediment transport can also be taken into account with the help of the shared stream-power model recently presented by

Hergarten (2020).

    Regarding the implementation in a large-scale landform evolution model, the main difference towards fluvial erosion is that glaciers are usually wider than the grid spacing in contrast to rivers. Including the finite width of glaciers is the main challenge in the implementation.

While the first model formulation assumes an infinitely thin ice layer, a finite thickness can also be included, where further approximations are necessary. With this extension, overdeepenings, hanging valleys, and steps at confluences can be simulated in an at least qualitatively reasonable way.

The implementation in the open-source landform evolution model OpenLEM uses the fully implicit scheme for erosion and sediment transport proposed by Hergarten (2020) in the context of fluvial landform evolution. This scheme allows for arbitrary time increments in principle, where changes in the flow pattern practically define an upper limit. The numerical effort is moderately higher than for the purely fluvial version and should be some orders of magnitude lower than for models based on the shallow-ice equations such as ICE-CASCADE and iSOSIA. Simulations even over several million years can be performed on standard PCs.

As a main limitation, the model presented here requires empirical relations for the width and the thickness of glaciers as a function of their ice flux. In contrast, models that implement the shallow-ice approximation directly are able to adjust the geometry of the cross section according to the initial geometry of the valley, its slope, and the parameters of the ice flux. A detailed benchmarking against the iSOSIA model as a reference with regard to the efficiency and to the question how well the stream-power-based model captures glacial erosion will be subject of a subsequent study.

*Code and data availability.* The open-source landform evolution model OpenLEM including the extensions presented here is freely available at http://hergarten.at/openlem. An additional package that contains all codes and simulated data is available at http://hergarten.at/openlem/esurf-2021-1.zip (preliminary location during the review phase). The author is happy to assist interested readers in reproducing the results and performing subsequent research.

*Author contributions.* N/A

*Competing interests.* The author declares that there is no conflict of interest.

*Acknowledgements.* The author would like to thank Eric Deal for his very constructive review and Wolfgang Schwanghart for the editorial handling.

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
