# Peer review of "Modeling glacial and fluvial landform evolution at large scales using a stream-power approach"

_Earth Surface Dynamics, 2021_

## Referee Comment (RC1)

Review of Hergarten 2021 for esurf

esurf-2021-1

I would just start my review by stating that when I was asked to review "A stream-power law for glacial erosion and its implementation in large-scale landform-evolution models" by Stefan Hergarten, I told the editor that I was happy to do the review, and felt that I had the relevant expertise, but also a conflict of interest. Although I do not know him personally, Dr. Hergarten and myself are coauthors on a recent paper on a topic very similar to the one dealt with in this paper.  Further, I have even more recently published a paper that has an overlapping focus to this one. However, although the topic matter between our two papers is very similar, the goal of each one is distinct, and therefore in my view the work is complementary. I informed the editor that I felt I was still able to maintain impartiality, though I wanted to be transparent about my relationship to Dr. Hergarten and his most recent work, and he agreed it was a minor conflict. The review that follows is unusual in that I discuss my own work extensively. I do so partly because it forms the basis of my understanding of the subject matter of this work. I also mention my work frequently because Dr. Hergarten had not evidently seen it before submitting this work, which is understandable due to how recent it is.  Due to the overlap between the two papers, my most recent paper is often very relevant, particularly to the first half of this manuscript. I hope this is not interpreted as a plug for my own work - in the end Dr. Hergarten is free to decide how much he wishes to use. It is rather an effort to arrive at the best possible topography-based model for glacial erosion.

In "A stream-power law for glacial erosion and its implementation in large-scale landform-evolution models" Dr. Hergarten presents the derivation of a topography-based model of glacial erosion in the spirit of the classic stream power incision model (SPIM) for fluvial erosion. After establishing the similarities between the SPIM and his new model for glaciers he proceeds to outline and solve the differences in the glacial model that are responsible for numerical challenges that do not exist for landscape evolution models (LEMs) based on the SPIM. This leads to a 2D LEM with coupled glacial and fluvial erosion and sediment transport which makes more assumptions than existing glacial LEMs, but also runs dramatically faster. This is important in opening up the ability to explore the parameter space of the model in a way that would not be possible with the more sophisticated models of glacial erosion.

There has been recent activity in developing simplified models of glacial erosion and taking advantage of decades of work with the SPIM to understand in a more quantitative way than ever before the role that glaciers play as agents of landscape evolution. In this I am clearly biased - but I find the work in this manuscript to be topical, relevant and important given this recent activity. Some of the key problems standing in the way of a 2D model of glacial erosion - in particular the implementation of a channel width that is generally larger than the model grid spacing - have been brought up by Dr. Hergarten, and solutions have been implemented. I find this already a significant contribution, but Dr. Hergarten has gone further, and implemented several potentially important processes, including sediment transport by glaciers and subglacial

fluvial erosion.

Though this manuscript is clearly novel, and constitutes an important contribution, I also have a few criticisms that would ideally be addressed before publication.

Dr. Hergarten is clearly very good with theoretical model development. However, one downside of that is that the theoretical development taking place in this paper far outstrips any empirical support. I don't find this in and of itself to be a critical problem - Dr. Hergarten has shown mathematically and then numerically how one would go about constructing a 2D LEM with glacial erosion. My criticism here rather lies with the packaging of the work. I think that there are many places where the limitations on the empirical understanding of the problems at hand should be made much more clear, and ideally discussed in more depth. Also the tone of the introduction could be modified to make it clear that this is more of a numerical implementation of yet to be developed - and tested - models of glacial erosion. In the same vein, I find there are many unsubstantiated comments made throughout the paper, which are based rather on an intuition for how glacial erosion works. I see where the intuition comes from, and I even often agree generally - but at the end of the day, even if these are my intutions, I would have to admit that I actually don't really know how the system 'should' behave. Intuition is important, particularly in model development. However, I think that it is important to maintain a standard of impartiality and try as far as possible to support intuition and intuition based statements with data and observations. That said, this work is rather new and there may often not be empirical data to cite - in this case however, there should be a higher level of transparancy about what behaviour has been observed and what is an educated guess. I have highlighted many of the places where I feel that more empirical support and/or circumspection would be valuable- but I also would encourage Dr. Hergarten to modify the tone throughout the paper to be more in line with an relatively untested theory of a physical process which we honestly don't understand very well yet. It may be a great way to identify where future empirical work could make the biggest impact for the development of good glacial erosion models.

At the end of the paper, I find that Dr. Hergarten has produced a convincing 2D model of glacial erosion that accounts for the most critical aspects of glaciers while retaining simplicity to keep the model fast. Future work may show us that some critical processes have been ignored, but I think that based on current knowledge, he's done a great job. However, there are several places throughout the paper that I disagree with the model development - particularly due to the fact that Dr. Hergarten temporarily makes assumptions that I don't think are appropriate, and does so without discussing how significant these assumptions are. My concern is that the critical elements that set glaciers apart from rivers are not appreciated. This seems detrimental to me for two reasons: readers will not appreciate the hardest aspects of the model development and where the focus of future work should be, and readers may think that any of the equations in the paper would be fair game for future work, when in fact I feel that only the 2D model is sufficiently complete to capture glacier erosion on landscape time and space scales. In particular, I think that the role of the ice surface slope versus channel slope and the ice accumulation rate (via some climate model with the ELA) need to be highlighted rather than somewhat implicitly being included in the 2D model. If Dr. Hergarten wishes to retain the current structure of the paper, then many of the equations shown - in particular 7-19 should be clearly described as intermediate steps which are not sufficient to describe glacial erosion until the assumptions of zero ice thickness and constant upstream ice production rate are removed in the 2D version of the model.

Finally, there are only a few empirical observations that can be currently used to test our theories of glacial erosion. One big one is U-shaped valleys. But the fact that valleys are parameterized in this model precludes this from being used to validate the model in any sense. One other observation that can be used is the observation and qualitative theory of the glacial buzzsaw - that very little terrain exists significantly above the ELA. There is also the associated, somewhat implicit conclusion that glacial terrain will have a different slope-uplift rate scaling. I urge Dr. Hergarten to take advantage of the speed of his new 2D model to compare more than just a few profiles of fluvial landscapes to glacial landscapes. It would be great to see a bit of an exploration of the parameter space - how does the overall channel slope or slope of the orogen change as a function of uplift rate or climate, and how is this different for purely fluvial landscapes compared to glacio-fluvial landscapes, or when sediment transport is turned on? Has Dr. Hergarten created a model that is fundamentally different that the SPIM, or do these landscapes look like fluvial landscapes with wider channels?

Overall I think this is a great piece of work that just needs some expansion, a bit more in depth explanation and a bit more polish. In line with my first main comment, there should be better citation of the literature to support comments made throughout the paper. It would be good to see a bit more careful handling of the assumptions made, and some more in depth analysis at the end of the paper.

Eric Deal

Detailed comments

Line 13: 'The difference in mathematical and numerical complexity may be the main reason for this imbalance' - This could be supported by some citations. Also I'm not sure I totally agree with it. When it comes to erosion, both processes are difficult to observe directly. It seems to me that one of the biggest reasons for the discrepency may be that rivers are much more prevalent. This makes them more important to study for many reasons important to society - particularly around river engineering projects. The same logic would also apply for landscape evolution models, where rivers are, in a global sense, much more important than glaciers. This could lead to a clear focus on fluvial erosion as the dominant process, resulting in better models of fluvial erosion available to be integrated into landscape evolution models. I think it is worth keeping in mind that nearly 100 years separates the first mention of erosion power being proportional to slope and discharge and the first time the stream power incision model was integrated into a landscape evolution model.

Line 15: While hack definitely mentioned transport capacity proportional to slope and area, wasn't this more in terms of gravel-bedded alluvial rivers? I feel like the standard first reference of Howard 1994 for the stream power incision model (SPIM) is used so because it was the first time that this was applied in the context of bedrock rivers, which would to me make it the correct reference for LEMs.

Line 19: I think you mean 'Lumped parameter K'

Line 21: 'more ore' > 'more or'

Line 21: Can you expand on the idea of universal a bit more? I'm not sure I understand what you mean by that. If, perhaps, you mean that they have universal values, I'm not sure that I agree with that. I know that the ratio of m to n is often observed to be within a narrow range, but the value of n in natural systems is highly variable, with observations commonly ranging from 2/3 to > 4. In any case, I feel like in addition to explaining this a bit more clearly, the statement needs to be supported with citations.

Line 26: This is an abrupt transition to 'fully implicit schemes'. Perhaps a new paragraph, as well as more context? I can guess you mean LEMs, but this is not very clear from the text. Why is implicit important? You state a few reasons, but the advantages of implicit over any alternatives is not clear to the uninitiated, and any drawbacks associated with implicit are not discussed. Given that there is a focus on retaining the ability to model this equation implicitly throughout the rest of the paper, I think it might be good to have a short discussion on implicit versus explicit schemes before moving on.

Line 37: There is some recent work on just this topic. Myself and Günther Prasicek recently published a paper which shows how a model of glacial erosion that is very comparable to the stream power incision model can be formulated where the erosion rate can be computed directly from topography and an ELA that plays the role equivalent to P in the SPIM. The analytic steady-state solution depends on an approximation of local ice surface slope as the average ice surface slope over the glacier (this doesn't really impact the steady state solutions however). But just to point out that it is possible, we have also written (but not published) a numerical solver without this last approximation, just using the upstream ice flux and then solving for local ice flux and ice surface slope which is stable and order n, though not implicit.
*Deal, Eric, and Günther Prasicek. "The Sliding Ice Incision Model: A new approach to understanding glacial landscape evolution." Geophysical Research Letters: e2020GL089263.*

Line 61: It is a bit more fair to say that the equation governing the relationship between ice flux, q and ice thickness h, is a 5th order polynomial with two terms, h^3 and h^5, and the coefficients in front of them (given the standard values from the literature - such as those taken from Prasicek et al, 2020) determine that the transition from sliding dominated to deformation dominated flux occurs at a thickness of a few hundred meters. This is right in the middle of the range expected in alpine glaciers, and makes it difficult to justify a sliding only approximation (or a deformation dominated approximation, unfortunately). I'm not saying that sliding only is a bad place to start, but I don't think that it is useful to discuss the ratio of vd to vs. That makes it seem like sliding only is an ok approximation. I think it's better to be clear about the fact that it is not a good approximation, but perhaps a mathematical necessity.

Line 66: I'm a bit confused about this concept of 'eliminating' h. I agree that there are two equations, and if h is the only relevant unknown, it can be considered redundant info, and does not have to appear in the equation. But this doesn't mean h doesn't exist. It can be calculated at any time using equation 4 or 7. Of course, it is clear to me that you know that. However, my confusion comes in later when it is discussed that this is the case with no thickness, and you will explicitly consider h from that point on. Perhaps I have really misunderstood the argument here - but I would argue that eliminating h from the relationship between vs and qi does not mean it is not explicitly accounted for, or that it can be considered 0. Instead I would argue that the place where you make h=0 is when you consider the relevant slope in the problem to be the bedrock slope instead of the channel slope (and perhaps to a lesser extent where you consider the elevation of precipitation to be the bedrock elevation instead of the bedrock plus ice thickness elevation). I think this distinction is important because it is the difference between the ice surface slope and the bedrock slope which is a big part of what makes this problem hard to solve from topography compared to the SPIM, and it is

also what makes the behaviour of the model different and interesting. I think it would be more informative and beneficial to the readers to make it clear that this is the crux of the problem, and that this is where you are making the key approximation for this first model. I would even encourage you to label the slope more explicitly, so it is clear when the equations are referring to bedrock slope and when they are referring to ice surface slope.

Line 70: I would argue, that since you have already implicitly stated that the goal is to recover a model of glacial erosion from topography (Line 37 and again in this line), that the task here is not just to describe w, but also to describe q in terms of A and some climate model like P does for the SPIM. This is the other hard part of the problem compared to the SPIM and should be mentioned here.

Line 75: I'm a little uncertain what a glacier polygon is - does a single polygon correspond to a full glacier - could this be expanded on a tiny bit?

Line 76: I appreciate the careful point made here about characteristic width versus actual width! However, I think that the conclusion 'it makes sense to follow the concept…' sort of undoes the care taken directly before it. Perhaps better would be something like "the proposition of Prasicek is not at odds with the observation of Bahr, though it cannot be concluded solely on the observations of Bahr. In the absence of any model that is better supported by data, and in order to stay in line with standard fluvial models, we use the model of prasicek." - this makes it clear that the model for channel width remains at this point unsupported by direct empirical observations (beyond those in the prasicek paper itself, which while they represent hard work, are, to be honest, not constraining any models of channel width).

Line 86: I think that there is an h in the equation where it should be eta. Also, not a big deal but I would maybe suggest a symbol for the width exponent that is not chi - that has taken on a pretty specific meaning in this field.

Line 87: Assuming a constant rate of ice production over the entire upstream catchment is a very significant assumption. I would argue that perhaps the defining characteristic of a glacier versus a river, at the landscape scale, is the fact that q is not proportional to A but in fact a convolution of a climate and the topography. This is technically also true in the fluvial world, but that turns out to be a second order detail. We know, or at least strongly suspect, that this is a first order feature of glaciers. Why else do glaciars have ELAs, termini, terminal moraines and long skinny lakes far below the snowline left strewn about the landscape as a reminder of their past extents. I actually don't think that this has a huge impact on equation 12/13, but the statement adds to the confusion that comes later with using Ai. I feel that it is still possible to hide all the complicated interaction between climate and topography in Ai - but this is not really made clear in the text. The use of A in Ai is also a bit confusing, because it makes it seem like it could be as simple as A ~ x^(1/eta), especially when combined with the statement of assuming a constant rate of ice production. It's not really clear for how long the assumption of constant ice production holds throughout the text. Does it also apply to equation 17, in which case Ai really is similar to A. However, in this case the most interesting and perhaps important feature of glaciers is missed - that they melt.  I want to say, I really appreciate where this work ends up, and I find the numerical model development to be important work. I also understand why you structured the paper the way you did. However, I strongly disagree with the way that this assumption, alongside the not-really-mentioned approximation that bedrock slope S can be used in place of ice surface slope are introduced without much discussion or justification. The path to equations 18/19 is made to seem simple, even inevitable. However, this is only the case when these two assumptions can be made, and these two assumptions are not really fair to our understanding of the workings of glaciers. It is my concern that readers may think that equations

17/18/19 would constitute a functional model of glacial erosion. This would only be true if there are in fact a lot of physics hidden in Ai - but the need for this is not made clear when the equations are introduced. I urge you to remove this statement entirely. You could, for example, take a path similar to the one that we took. This was simply to point out that models of channel width for glaciers are poorly constrained, and it anyways makes more physical sense that the channel width is proportional to ice flux rather than contributing area - this is because for glaciers there is no connection between contributing area and ice flux due to melting. Therefore, we simply propose equation 13 as an alternative to equation 12 and move forward with that. There is no need to invoke the fairly limiting assumption that there is a constant rate of ice production over the entire upstream catchment.

Line 93: This is a cool result to be sure! If we apply the sliding only approximation to our equations (gamma = 1) then we recover the same values for the exponents as you have here, which is also cool, though we always consider S to be ice surface slope. We found that this approximation lead to fairly substantial misfit between the steady state morphology of a glacio-fluvial profile when compared to the model presented by Prasicek 2020, which has fewer approximations than either topography based model of glacial erosion (yours or ours). However, we use an approximation for mixed sliding and deformation that does not increase the mathematical complexity of the model, yet leads to almost no misfit at steady state between the more complete model of Prasicek 2020 and our topography based model of glacial erosion. We stronly encourage you to use it as it increases accuracy without increasing the difficulty of solving the model. One caveat: we have carefully tested the accuracy of the various approximations at steady-state, but cannot attest to how well the accuracy holds during transient conditions. Very likely the approximate, topography based model exhibits some misfit during transience, but we would still expect the mixed sliding/deformation approx (gamma=2/3) to perform better than either sliding only or deformation only approximation.

line 97: Would be nice to have a citation supporting the statement about psi.

Line 99: Can you expand on what exactly this implies for the factor of proportionality in 14? I'm not sure I followed that very well.

Line 100: I agree, of course that this is the main difference between 2 and 14, but I don't agree with the reason. As you yourself have mentioned, there is a significant advantage to a topographic based model of erosion, and this is likely the real reason that 2 is written in terms of A and not Q. I think that this is an important distinction, because it had to be demonstrated for fluvial landscapes that $Q \sim A^p$, where p is close to 1. This jump will, unfortunately, not be so easy for glaciers.

equation 17: I understand the logic behind this equation, and obviously it's mathematically sound. However, I strongly feel that using a variable termed Ai is confusing and maybe a little misleading. It hides the complexity of the relationship between landscape and climate in a glacier network. A parameter 'po' can of course be defined, but what is the physical significance of it? In our work, we also had the same urge as you have here - to show the similarity between topography-based models of glacial and fluvial erosion. In the end we chose to define qi = IA, which is analogous to the fluvial equation q = PA. The difference between this approach and 16/17 is that I is undeniably a function of position, where P can be considered a function of position, but can also be approximated as a constant - an approximation that has been tested and shown to be not terrible. The other difference is that A is A with no distinction between fluvial or glacial and therefore $qi \sim Ix^{(1/eta)}$. However, this means that the closest one can come to the SPIM is $E = K\_g((I(x)A(x))^{\theta} S\_{\text{ice surface}}(x))^{\ell}$. I would strongly encourage you to not use Ai, because it is confusing. Since Ai is effectively unknowable given the setup here, it is also not really a topography-based erosion model at this point

anyways. I think you should either stop at qi, and point out that the SPIM can and is often written this way, or adopt an approach similar to the one that we did in Deal and Prasicek 2020.

Equation 19: I think at some point between equation 4 and 19 it needs to be clearly and visibly pointed out that the ice surface slope has been exchanged for the channel slope. The thickness of glaciers is within an order of magnitude of the height of the topography, and therefore the slope of the ice surface and the slope of the channel do not have to match at all - it is even standard for them to have opposite signs as the terminus of the glacier is approached: 1.Alley, R. B., Lawson, D. E., Larson, G. J., Evenson, E. B. & Baker, G. S. Stabilizing feedbacks in glacier-bed erosion. Nature 424, 758–760 (2003).

Equations 20 and 21: I strongly disagree with the use of the symbols A and Ai here. I do get the equations - the math is solid, it all checks out. There is nothing fundamentally wrong with the math here. However, I feel like these equations result from a little mathematical gymnastics that hinder the intuitive understanding of the model. It seems to me what's being shown here is actually fundamentally q, normalized by a constant po. We could write 20 as

$$A = q / po, \text{ where } q = s*p + sum\_over\_donors(q)).$$

The constant po does allow this to be called A, but particularly in the glacial case where po is effectively unknowable a priori, this is not a helpful equality. I think it would be easier to follow and more properly understood if it were referred to as q. One argument to back this up is figure 5: A cannot decrease downstream. That's not how a concentrative network can function. However, Q can and does decrease downstream in some real rivers and most real glaciers. The other argument is that A is a topographic parameter - it should be calculated from the geometry of the landscape with no concern for things such as precipitation rate. The need to include p, pi and po in equations 20 and 21 is a giveaway that they are not really calculating area.

Line 134: 'over an area around to the cardinal flow path" - wording is a bit awkward, can drop the 'to'.

Line 136: I think you mean equation 13

Line 137: Change 'prolonging' to 'extending'

Line 137: This is clearly a place where no standard has been has been proposed or accepted, so it is exciting to consider the possibilities. I feel that you dismiss these two cases perhaps too rapidly. A couple things come up for me - what makes them endmember cases? It's not clear to me on what spectrum? What variable is reaching an end value? Also what makes them unrealistic? It seems to me that with single timesteps of potentially hundreds to thousands of years converging immediately to U-shaped valleys is not necessarily immediate or unrealistic. This would be the analagous case to the SPIM, where the channel shape cannot evolve, but is immediately specified and whose erosion rate is constant across the entire channel. I understand that the situation is different here because channels are no longer subgrid, but still this seems to me one of the more reasonable choices. Two questions come up for me when thinking about how to handle channel width. First, even though channels are no longer subgrid, is there really enough spatial resolution in the channel that modeling nonuniform erosion rates across channels would be at all realistic? Second, is it worth the effort? What effect would nonuniform channel erosion or non U-shaped valleys have on the largescale evolution of the landscape? I can imagine that right as the channel network develops these dynamics could play a role, but after the network topology is mostly established (e.g. just after the first few timesteps - long before steady state) does it really impact ice routing or erosion rate at a scale greater than say a few channel widths?

Line 140: change 'a prolongation of the' to 'extending the'

Line 142: Again, I am guessing the final handling of channel width is not critical for the behaviour of the model - however, I don't agree that extending the ice flux Ai to the channel edges is the most realistic approach. If I have understood correctly, the parameter Ai is the volumetric flux of ice in the entire channel - the idea of extending it to the edge is for me a bit confusing. Technically there is only one Ai for the entire channel width at any give location - no extending needed. However, for me to make sense of it I have to go back to the original definition of the erosion rate - a function of the sliding velocity. Therefore the most realistic would be to extend to sliding velocity across the channel - either say that the sliding velocity is constant across the channel, or that there is a known gradient. You have constructed a relationship between volumetric flux and flow depth that then leads to a sliding velocity that was maybe implicitly meant to be the centerline - though this was never stated. Then perhaps in this world, the most realistic would be to recongnize that ice won't really be able to sustain topograpy on its surface, and the elevation of the ice will be close to flat across the channel - then the model valley could be filled with ice until the value Ai is used up, and the erosion rates calculated from the ice surface slopes across the channel and the flow depth across the channel. However what I think all this messing about with across channel erosion rate is really saying is that we don't have enough ice flow physics in this model to calculate across channel erosion rates. I feel that we have seen that the U-shaped valley is a sort of equilibrium channel form where the erosion rate can be constant across the channel. This is supported I would argue implicitly by the existence of U-shaped valleys in all glacial landscapes around the earth as well as some modeling studies (Leith, K., Moore, J. R., Amann, F. & Loew, S. Subglacial extensional fracture development and implications for Alpine Valley evolution. J Geophys Res Earth Surf 119, 62–81 (2014).). Accepting this channel form a priori as an equilibrium form similar to the way we accept the channel width-discharge relationship for the SPIM leads to the conclusion, for me, to acknowledge that modelling the evolution of channel width is beyond the capabilites of this model and that the best approach is to extend the erosion rate of the centerline to the channel edges.

Line 157: I feel that the usage of the value of 1 or 2 for most of the key parameters is strange. This plays the role of a nondimensionalization, but is much harder to follow. For example, K = 1 and U = 1 for n = 1, whereas the 'standard' values (K = 1e-6, U = 1e-3 for n = 1) give a U/K ratio of 1e3 rather than 1. Probably this doesn't matter, and the landscape can just be vertically scaled - but then I, as the reader, have to do all that thinking about it. I must try to interpret the model parameters in terms I am familiar with, and the whole time I am wondering if that dramatically different U/K ratio really doesn't matter at all. I feel that this is mental effort more effectively conducted by the author, and I would really appreciate either a proper nondimensionalization, or simply the usage of more familiar parameter values.

Equation 22: what is A here? Is it just s + sum_over_doners(s) or is it equation 20? If it is equation 20, why does p come into Ai,eff?

Equation 22: I think you need to go into a fair bit more detail about how you arrived at this equation. I feel that this is an important point, but I don't understand why A comes into the ice flux. This seems to me like it would cause the ice to not melt fast enough below the ELA because there would be a portion of the ice flux that is always going to increase downstream regardless of elevation. Also I can't understand at all why the valley shape would change as a function of Ai,eff - can you provide some insight into that?

Line 174: I think you mean equation 13 again...

Line 208: Why is Kg 2K? Why is it larger, and why 2 times larger specifically? I feel that this statement implies

conclusions that have already been made about glacial erosion being more erosive than rivers. Why not just leave Kg = 1 in the absence of any specific knowledge about Kg? Otherwise one could argue that it should actually be 3, or 10 or 100… Alternatively, there are empirical estimates of Kg out there.

Figure 5: It should be specified clearly that x is distance from outlet of river.

Line 224: I think that the statement about the glacial profiles being steeper is fairly disingenious. It is contingent on choosing the right elevation as your reference point. In fact, across the whole profile the glaciers are actually less steep. Also much of the glacial profile is less steep, the only reason that they are steeper for given reference elevations is because of the steep steps in them. As you state yourself, the only reason there is a step is because of the ice dynamics at the terminus - so the second step in those profiles is actually a transient feature! Comparing transient fluvial profiles and concluding something about their erosion efficiency would be considered a fairly flawed analysis. Also to be fair, even the first step is partly a function of having only sliding and partly a function of not considering the ice surface slope. From what we've seen in our models, there is sometimes a step, and often no step - and when the sliding only condition is imposed, there is much more likely to be a step. Also there doesn't need to be a step at all because it is actually the ice surface slope that drives erosion rate, and while it is true that the ice surface slope goes to infinity, this results in a rapid thinning of the glacier, which often manifests in a channel slope with a sign opposite to the that of the ice surface slope - and therefore no step. Finally, while this step may be partly the result of unrealistic model conditions (that ice flux goes to zero) , this can be handled with a harmless solution like saying that the glacier ends when the ice is just a few meters thick, instead of waiting for it to go to zero. Similar to how the linear stream power law implies infinite elevation, but this is clearly not the case.

Line 226: Also, while I agree that requiring infinite ice surface slope at the glacial terminus is unrealistic, I am not convinced that the steps are unrealistic. I think this would be a good place to support this statement with empirical evidence. I would argue that steps / steep fronts under glaciers or at glacial termini are actually quite common - and how can we (yet) say where they come from?

[Figure]

[Figure]

Line 227: I also agree that erosion by meltwater is important - but you hardly need my agreement for that - there is plenty of evidence, outside of the existence/non-existence of steep fronts to support this conclusion. I think it would be good to cite that here.

Line 231: I like the idea of this, but can you do better than assuming? There is work on subglacial channel erosion, can some of it be cited here to show that it is at least reasonable to claim an equation of the form of 25?

equation 26: Seems like there is an i subscript missing here on the first A

Lines 230 - 245: I like the simplicity of these equations, but I think you need to be more open about the fact that at this point, equation 27 is wildly unconstrained. Can one even begin to put somewhat empirical values to these Ks? Either that or bring in some more empirical work on subglacial fluvial erosion.

Line 245: "Since there is no discontinuity in the erosion rate at the glacier terminus then, the changes in the flow pattern are much smaller than for the version without erosion by meltwater. The respective profiles depicted in Fig. 7 (dashed lines) reveal a smooth transition from the glacial regime to the fluvial regime". It is clear to me from the text that you find this to be preferable to the previous case with the steep front. I understand this intuition, but I feel obligated to ask why, precisely, is this better? There have been no observations brought out to show that this is really more realistic. How well studied are glacial channel profiles really? Can we really fairly say we already have an intuition for what they should look like? Keeping in mind also that the glacial channels we see today are far, far out of steady state due to us being in an interglacial period. I don't really feel that the section on subglacial fluvial erosion has to be removed or anything, but I do think that the validation of this process based on any attributes of the resulting profiles is potentially folly, and should be avoided. I think you should be more empirically motivated when assessing how realistic these profiles are.  It would be sufficient for the development of the model to point out that subglacial fluvial erosion clearly happens - and make some citations, and then, while being open about how unconstrained the parameter values are (unless it is possible to constrain them given existing theory and empirical observations), impartially discuss the differences in the resulting profiles. Or perhaps it is possible to show that the value of Kf is relatively unimportant for the dynamics, that would also be nice.

Figure 7/8 - can you confirm that the value of Kf is the same as K somewhere in the text or figure caption?

Line 256: I think in order to make a statement like this, you need some observations to support it.

Line 262: "Let us assume that the identical relations for glacial and fluvial erosion do not only hold for the detachment-limited end-member, and that the shared stream-power model provides a reasonable description of both glacial erosion and erosion by meltwater. " - This strikes me as a huge assumption. At this point we have not talked about glacial sediment transport at all. I know that for rivers, both models of erosion and sediment transport are based on the fluid shear stress at the bottom of the river, and even without

considering the decades that have gone into studying these proceses, there are good reasons to think that they would be related to some degree. Is this the case with glaciers? To be honest, I don't really know how glacial sediment transport works, so I can't comment on it - but it would be, I think, important to see some empirically based arguments, even if they are fairly hand-wavey, as to why we might believe that glacial transport would also be proportional to KA^(m+1)S^n.

Line 321: I'm not sure if this is a reasonable justification. It is akin to saying that if E = KA^mS^n would lead to no steady state when slope goes to zero, so elevation will grow out of control. The answer is not to remove the slope dependence of erosion rate, but to recognize the limits of the model, and that a purely detachment driven model with no explicit fluid dynamics will never find a slope that is zero at steady state.

Section 6: I think it would be important to see some sort of comparison between the non-slope dependent ice thickness you implement in the implicit versin of the model and the real ice thickness implemented in an explicit version of the model to show that the approximation is reasonable and recovers an ok answer most of the time.

---

## Author Response (AR1)

Dear Reviewers, dear Editor,

I would particularly thank Eric Deal for his constructive and encouraging comments, although it would not be possible to take all comments into account without transferring the authorship to him.

I already expected that the glaciology community would be very critical towards simplifying the models instead of going deeper into the details of the involved processes. However, the community comment of Flavien Beaud – written in the form of a review – was a bit disappointing for me. I definitely do not want to raise any doubts on his expertise, but his arguments are mainly on the level of keywords. I have experienced this style from old researchers, but I was surprised to see this way of tearing down others' work without considering it seriously from an active, promising researcher.

In this sense, I found the comments of Marc Jaffrey even more problematic. The first version of his review consisted mainly of general phrases. The second version (which I refer to below) was somewhat more precise, but still of low quality from my point of view. In particular, the mathematical aspects are totally wrong raise some doubts on the qualification of the second reviewer. Apart from this, the reviewer refers to opinions of the glaciology community, but unfortunately I did not find any documented contributions of the reviewer to this field, except for a conference poster. So I have serious doubts about not only on the mathematical background of the second reviewer, but also on his experience in publishing and reviewing.

In the following, the points addressed in the two reports are discussed, and changes to the manuscript are described. Line numbers refer to the version with highlighted changes at the end of this document.

**Reviewer 1 (Eric Deal)**

I would just start my review by stating that when I was asked to review "A stream-power law for glacial erosion and its implementation in large-scale landform-evolution models" by Stefan Hergarten, I told the editor that I was happy to do the review, and felt that I had the relevant expertise, but also a conflict of interest. Although I do not know him personally, Dr. Hergarten and myself are coauthors on a recent paper on a topic very similar to the one dealt with in this paper. Further, I have even more recently published a paper that has an overlapping focus to this one. However, although the topic matter between our two papers is very similar, the goal of each one is distinct, and therefore in my view the work is complementary. I informed the editor that I felt I was still able to maintain impartiality, though I wanted to be transparent about my relationship to Dr. Hergarten and his most recent work, and he agreed it was a minor conflict.

I fully agree that there is definitely no conflict of interest, although we somehow worked in similar directions after we were coauthors of the paper of Günther Prasicek in EPSL.

The review that follows is unusual in that I discuss my own work extensively. I do so partly because it forms the basis of my understanding of the subject matter of this work. I also mention my work frequently because Dr. Hergarten had not evidently seen it before submitting this work, which is understandable due to how recent it is. Due to the overlap between the two papers, my most recent paper is often very relevant, particularly to the first half of this manuscript. I hope this is not interpreted as a plug for my own work – in the end Dr. Hergarten is free to decide how much he wishes to use. It is rather an effort to arrive at the best possible topography-based model for glacial erosion.

In "A stream-power law for glacial erosion and its implementation in large-scale landform-evolution models" Dr. Hergarten presents the derivation of a topography-based model of glacial erosion in the spirit of the classic stream power incision model (SPIM) for fluvial erosion. After establishing the similarities between the SPIM and his new model for glaciers he proceeds to outline and solve the differences in the glacial model that are responsible for numerical challenges that do not exist for landscape evolution models (LEMs) based on the SPIM. This leads to a 2D LEM with coupled glacial and fluvial erosion and sediment transport which makes more assumptions than existing glacial LEMs, but also runs dramatically faster. This is important in opening up the ability to explore the parameter space of the model in a way that would not be possible with the more sophisticated models of glacial erosion.

There has been recent activity in developing simplified models of glacial erosion and taking advantage of decades of work with the SPIM to understand in a more quantitative way than ever before the role that glaciers play as agents of landscape evolution. In this I am clearly biased – but I find the work in this manuscript to be topical, relevant and important given this recent activity. Some of the key problems standing in the way of a 2D model of glacial erosion – in particular the implementation of a channel width that is generally larger than the model grid spacing – have been brought up by Dr. Hergarten, and solutions have been implemented. I find this already a significant contribution, but Dr. Hergarten has gone further, and implemented several potentially important processes, including sediment transport by glaciers and subglacial fluvial erosion.

I was indeed not aware that you and Günther Prasicek worked in a similar direction and missed your paper in GRL. I was even a bit disappointed to see that you were definitely faster than me with the first part of the theory, presumably because I spent much work on the numerical implementation. The fact that you are already familiar with the first part of the theory, however, makes it a bit difficult for me to assess those points where you misunderstood my concept. It is difficult to see whether this happened although you are already familiar with some of the ideas or even particularly because you are, while using a different formalism or a slightly different idea.

Though this manuscript is clearly novel, and constitutes an important contribution, I also have a few criticisms that would ideally be addressed before publication.

Dr. Hergarten is clearly very good with theoretical model development. However, one downside of that is that the theoretical development taking place in this paper far outstrips any empirical support. I don't find this in and of itself to be a critical problem – Dr. Hergarten has shown mathematically and then numerically how one would go about constructing a 2D LEM with glacial erosion. My criticism here rather lies with the packaging of the work. I think that there are many places where the limitations on the empirical understanding of the problems at hand should be made much more clear, and ideally discussed in more depth. Also the tone of the introduction could be modified to make it clear that this is more of a numerical implementation of yet to be developed – and tested – models of glacial erosion. In the same vein, I find there are many unsubstantiated comments made throughout the paper, which are based rather on an intuition for how glacial erosion works. I see where the intuition comes from, and I even often agree generally – but at the end of the day, even if these are my intuitions, I would have to admit that I actually don't really know how the system should' behave. Intuition is important, particularly in model development. However, I think that it is important to maintain a standard of impartiality and try as far as possible to support intuition and intuition based statements with data and observations. That said, this work is rather new and there may often not be empirical data to cite – in this case however, there should be a higher level of transparency about what behaviour has been observed and what is an educated guess. I have highlighted many of the places where I feel that more empirical support and/or circumspection would be valuable – but I also would encourage Dr. Hergarten to modify the tone throughout the paper to be more in line with an relatively untested theory of a physical process which we honestly don't understand very well yet. It may be a great way to identify where future empirical work could make the biggest impact for the development of good glacial erosion models.

I thought quite much about the structure of the paper and arrived at a version starting from an oversimplified version where many aspects are missing, but where there is at least some solid ground. Then extensions such as meltwater, sediment transport, and finally finite ice thickness are introduced step by step. It is clear that these extensions successively leave the solid ground. So I think this step-by-step approach is the only way to keep the model as a whole open for improvements in the individual components. I know that parts of the community would prefer a classical methods section just collecting equations and parameter values and then run some simulations, but I do not like this so much.

The question whether an approach or an equation is intuitively correct is indeed more important for me than justifying it formally by citations, potentially (but unfortunately even quite often) taken from a different context, although I accept that this is the way large parts of research are proceeding nowadays.

Anyway, the static parameterization of the glacier width by the ice flux is probably the most critical approximation, in particular if we leave the world of steady-state longitudinal profiles. As you used the same approximation in your paper, it obviously appears to be less critical to you than the approaches introduced later, but I would not agree here.

At the end of the paper, I find that Dr. Hergarten has produced a convincing 2D model of glacial erosion that accounts for the most critical aspects of glaciers while retaining simplicity to keep the model fast. Future work may show us that some critical processes have been ignored, but I think that based on current knowledge, he's done a great job. However, there are several places throughout the paper that I disagree with the model development – particularly due to the fact that Dr. Hergarten temporarily makes assumptions that I don't think are appropriate, and does so without discussing how significant these assumptions are. My concern is that the critical elements that set glaciers apart from rivers are not appreciated. This seems detrimental to me for two reasons: readers will not appreciate the hardest aspects of the model development and where the focus of future work should be, and readers may think that any of the equations in the paper would be fair game for future work, when in fact I feel that only the 2D model is sufficiently complete to capture glacier erosion on landscape time and space scales. In particular, I think that the role of the ice surface slope versus channel slope and the ice accumulation rate (via some climate model with the ELA) need to be highlighted rather than somewhat implicitly being included in the 2D model. If Dr. Hergarten wishes to retain the current structure of the paper, then many of the equations shown – in particular 7-19 should be clearly described as intermediate steps which are not sufficient to describe glacial erosion until the assumptions of zero ice thickness and constant upstream ice production rate are removed in the 2D version of the model.

In my opinion, these points are mainly interpretations of approximations that are in fact not made. **I tried to clarify these aspects (see detailed points).**

Finally, there are only a few empirical observations that can be currently used to test our theories of glacial erosion. One big one is U-shaped valleys. But the fact that valleys are parameterized in this model precludes this from being used to validate the model in any sense. One other observation that can be used is the observation and qualitative theory of the glacial buzzsaw – that very little terrain exists significantly above the ELA. There is also the associated, somewhat implicit conclusion that glacial terrain will have a different slope-uplift rate scaling. . . .

Validation is indeed hampered by the direct parameterization of glacial features. This mainly concerns the width and the thickness of the ice layer. So V-shaped valleys occur as well as hanging valleys and overdeepening. Quantitatively, these are rather directly related to model parameters. The same applies to the glacial erodibility, which can in principle be adjusted from almost fluvial to an extreme buzzsaw effect. In this sense, the predictive power of the model is indeed lower than that of other models. But to be honest, the predictive power of other models is also not extremely high due to a huge number of parameters that are not well constrained.

...I urge Dr. Hergarten to take advantage of the speed of his new 2D model to compare more than just a few profiles of fluvial landscapes to glacial landscapes. It would be great to see a bit of an exploration of the parameter space – how does the overall channel slope or slope of the orogen change as a function of uplift rate or climate, and how is this different for purely fluvial landscapes compared to glacio-fluvial landscapes, or when sediment transport is turned on? ...

Yes, of course – but not in this paper. There would be so many aspects to be considered that it will immediately become twice as long as it already is.

...Has Dr. Hergarten created a model that is fundamentally different that the SPIM, or do these landscapes look like fluvial landscapes with wider channels?

Wide, U-shaped fluvial valleys with overdeepenings, and confluence steps, and with a rather complex relation between flux and river length.

Overall I think this is a great piece of work that just needs some expansion, a bit more in depth explanation and a bit more polish. In line with my first main comment, there should be better citation of the literature to support comments made throughout the paper. It would be good to see a bit more careful handling of the assumptions made, and some more in depth analysis at the end of the paper.

Detailed comments

Line 13: 'The difference in mathematical and numerical complexity may be the main reason for this imbalance' – This could be supported by some citations. Also I'm not sure I totally agree with it. When it comes to erosion, both processes are difficult to observe directly. It seems to me that one of the biggest reasons for the discrepancy may be that rivers are much more prevalent. This makes them more important to study for many reasons important to society – particularly around river engineering projects. The same logic would also apply for landscape evolution models, where rivers are, in a global sense, much more important than glaciers. This could lead to a clear focus on fluvial erosion as the dominant process, resulting in better models of fluvial erosion available to be integrated into landscape evolution models. I think it is worth keeping in mind that nearly 100 years separates the first mention of erosion power being proportional to slope and discharge and the first time the stream power incision model was integrated into a landscape evolution model.

The engineering projects and the greater relevance to society are good points since glacial erosion was also studied scientifically in the 19th century. **I expanded this aspect a bit (lines 12–19).**

Line 15: While Hack definitely mentioned transport capacity proportional to slope and area, wasn't this more in terms of gravel-bedded alluvial rivers? I feel like the standard first reference of Howard 1994 for the stream power incision model (SPIM) is used so because it was the first time that this was applied in the context of bedrock rivers, which would to me make it the correct reference for LEMs.

Difficult to say – when searching for the rivers considered by Hack in Google Earth years ago, I got the impression that they are very different. This is why I wrote it in this somewhat unspecific way just mentioning where the relation formally occurred for the first time. I find it a bit difficult to follow it back in landform evolution modeling since the models of optimal channel networks around 1990 were also quite close to this. However, I agree that Howard was the first to use such a relation directly in a landform evolution model and **added the reference here (line 22)**.

Line 19: I think you mean 'Lumped parameter K'.

**Fixed, thanks (line 26)!**

Line 21: 'more ore' → 'more or'

**Fixed, thanks (line 28)!**

Line 21: Can you expand on the idea of universal a bit more? I'm not sure I understand what you mean by that. If, perhaps, you mean that they have universal values, I'm not sure that I agree with that. I know that the ratio of m to n is often observed to be within a narrow range, but the value of n in natural systems is highly variable, with observations commonly ranging from 2/3 to > 4. In any case, I feel like in addition to explaining this a bit more clearly, the statement needs to be supported with citations.

**I changed the text a bit (line 27)** in order to clarify that I meant what you suspect and where you do not fully agree. My personal point of view is that most of the older studies assumed that these values are indeed universal and mainly struggled about these values themselves (particularly $n > 1$ or $n < 1$). There is at least one study (Harel et al. 2016) that found an extreme scatter in the values and also apparent relations to climate. However, after investigating the data supplement of that study, I trust not too much in the results. Overall, I am quite sure that many studies in this field based on the SPIM are biased by unresolved effects of sediment transport. However, this discussion would drift off from the streamline of this study, so I prefer not to open it here and leave it in the form "more or less universal".

Line 26: This is an abrupt transition to 'fully implicit schemes'. Perhaps a new paragraph, as well as more context? I can guess you mean LEMs, but this is not very clear from the text. Why is implicit important? You state a few reasons, but the advantages of implicit over any alternatives is not clear to the uninitiated, and any drawbacks associated with implicit are not discussed. Given that there is a focus on retaining the ability to model this equation implicitly throughout the rest of the paper, I think it might be good to have a short discussion on implicit versus explicit schemes before moving on.

**I expanded this part a bit and moved it to lines 45–51.** Concerning the drawbacks of implicit schemes, I guess you suggest to emphasize that practically no implicit schemes of higher order exist for the advection equation. This would, however, get too deep into details here since we would need to discuss how serious the numerical diffusion of the first-order scheme really is here.

Line 37: There is some recent work on just this topic. Myself and Günther Prasicek recently published a paper which shows how a model of glacial erosion that is very comparable to the stream power incision model can be formulated where the erosion rate can be computed directly from topography and an ELA that plays the role equivalent to P in the SPIM. The analytic steady-state solution depends on an approximation of local ice surface slope as the average ice surface slope over the glacier (this doesn't really impact the steady state solutions however). But just to point out that it is possible, we have also written (but not published) a numerical solver without this last approximation, just using the upstream ice flux and then solving for local ice flux and ice surface slope which is stable and order $n$, though not implicit. Deal, Eric, and Günther Prasicek. "The Sliding Ice Incision Model: A new approach to understanding glacial landscape evolution." Geophysical Research Letters: e2020GL089263.

I was indeed not aware of this paper and – of course – **included it now and compared my approach to your model at some occasions (lines 91–95, 141–151, 503–504, 587–589).** Nevertheless, the statement written here is still true since your approach (as well as mine) still requires the slope of the ice surface. **In addition, I adjusted the title of the paper in order not to pretend that my glacial stream-power law was entirely new.**

Line 61: It is a bit more fair to say that the equation governing the relationship between ice flux, q and ice thickness h, is a 5th order polynomial with two terms, $h^3$ and $h^5$, and the coefficients in front of them (given the standard values from the literature - such as those taken from Prasicek et al, 2020) determine that the transition from sliding dominated to deformation dominated flux occurs at a thickness of a few hundred meters. This is right in the middle of the range expected in alpine glaciers, and makes it difficult to justify a sliding only approximation (or a deformation dominated approximation, unfortunately). I'm not saying that sliding only is a bad place to start, but I don't think that it is useful to discuss the ratio of vd to vs. That makes it seem like sliding only is an ok approximation. I think it's better to be clear about the fact that it is not a good approximation, but perhaps a mathematical necessity.

**I now mentioned your approach in the beginning of the section and explained the difference concerning the ice thickness (lines 91–95), why I think that starting from the zero-thickness approximation also has an advantage and how it will be addressed at the end of the paper (lines 141–151).**

Line 66: I'm a bit confused about this concept of 'eliminating' h. I agree that there are two equations, and if h is the only relevant unknown, it can be considered redundant info, and does not have to appear in the equation. But this doesn't mean h doesn't exist. It can be calculated at any time using equation 4 or 7. Of course, it is clear to me that you know that. However, my confusion comes in later when it is discussed that this is the case with no thickness, and you will explicitly consider h from that point on. Perhaps I have really misunderstood the argument here - but I would argue that eliminating h from the relationship between vs and qi does not mean it is not explicitly accounted for, or that it can be considered 0. Instead I would argue that the place where you make h=0 is when you consider the relevant slope in the problem to be the bedrock slope instead of the channel slope (and perhaps to a lesser extent where you consider the elevation of precipitation to be the bedrock elevation instead of the bedrock plus ice thickness elevation). I think this distinction is important because it is the difference between the ice surface slope and the bedrock slope which is a big part of what makes this problem hard to solve from topography compared to the SPIM, and it is also what makes the behaviour of the model different and interesting. I think it would be more informative and beneficial to the readers to make it clear that this is the crux of the problem, and that this is where you are making the key approximation for this first model. I would even encourage you to label the slope more explicitly, so it is clear when the equations are referring to bedrock slope and when they are referring to ice surface slope.

Sorry, but eliminating a variable from a system of equations always means that you remove it formally by combining the equations. I cannot imagine that any reader could think that $h$ does not exist any more afterwards. And at this point it should be clear from the previous equations that $S$ is the slope of the ice surface (line 59). Approximating this slope by the slope of the bed is part of Sect. 3, so I think it is neither helpful to address this here once more nor to introduce more symbols.

Line 70: I would argue, that since you have already implicitly stated that the goal is to recover a model of glacial erosion from topography (Line 37 and again in this line), that the task here is not just to describe w, but also to describe q in terms of A and some climate model like P does for the SPIM. This is the other hard part of the problem compared to the SPIM and should be mentioned here.

While this is true, I wrote it explicitly like this in order to point out that the estimate of $w$ is immediately needed without implying that this is sufficient. But beyond this, a stream-power law in terms of flux would already be similar to a stream-power law in terms of catchment size. Apart from this, I feel that opening too many side branches is not necessarily helpful for the readers.

Line 75: I'm a little uncertain what a glacier polygon is - does a single polygon correspond to a full glacier - could this be expanded on a tiny bit?

Yes, this is the case, but for some glaciers, multiple polygons referring to different times are available. But wouldn't it have been better to address this question to Günther Prasicek when you co-authored the 2020 EPSL paper?

Line 76: I appreciate the careful point made here about characteristic width versus actual width! However, I think that the conclusion 'it makes sense to follow the concept...' sort of undoes the care taken directly before it. Perhaps better would be something like 'the proposition of Prasicek is not at odds with the observation of Bahr, though it cannot be concluded solely on the observations of Bahr. In the absence of any model that is better supported by data, and in order to stay in line with standard fluvial models, we use the model of Prasicek.' - this makes it clear that the model for channel width remains at this point unsupported by direct empirical observations (beyond those in the Prasicek paper itself, which while they represent hard work, are, to be honest, not constraining any models of channel width).

Good point, **I expanded it accordingly (lines 113–116).**

Line 86: I think that there is an h in the equation where it should be $\eta$. Also, not a big deal but I would maybe suggest a symbol for the width exponent that is not $\chi$ – that has taken on a pretty specific meaning in this field.

Indeed, thanks! **I fixed it (line 124) and also replaced $\chi$ by $\xi$ (lines 108–110).**

Line 87: Assuming a constant rate of ice production over the entire upstream catchment is a very significant assumption. I would argue that perhaps the defining characteristic of a glacier versus a river, at the landscape scale, is the fact that q is not proportional to A but in fact a convolution of a climate and the topography. This is technically also true in the fluvial world, but that turns out to be a second order detail. We know, or at least strongly suspect, that this is a first order feature of glaciers. Why else do glaciers have ELAs, termini, terminal moraines and long skinny lakes far below the snowline left strewn about the landscape as a reminder of their past extents. I actually don't think that this has a huge impact on equation 12/13, but the statement adds to the confusion that comes later with using Ai. I feel that it is still possible to hide all the complicated interaction between climate and topography in Ai - but this is not really made clear in the text. The use of A in Ai is also a bit confusing, because it makes it seem like it could be as simple as $A \sim x^{1/\eta}$, especially when combined with the statement of assuming a constant rate of ice production. It's not really clear for how long the assumption of constant ice production holds throughout the text. Does it also apply to equation 17, in which case Ai really is similar to A. However, in this case the most interesting and perhaps important feature of glaciers is missed - that they melt. I want to say, I really appreciate where this work ends up, and I find the numerical model development to be important work. I also understand why you structured the paper the way you did. . . .

The constant rate of ice production is – of course – only used as a justification for assuming that the width depends on the ice flux instead of the catchment size, so for the step from Eq. (12) to Eq. (13). As you state, all the rest would not make much sense if this assumption persisted. **I pointed out this more clearly now (lines 125–133).**

. . . However, I strongly disagree with the way that this assumption, alongside the not-really-mentioned approximation that bedrock slope S can be used in place of ice surface slope are introduced without much discussion or justification. The path to equations 18/19 is made to seem simple, even inevitable. However, this is only the case when these two assumptions can be made, and these two assumptions are not really fair to our understanding of the workings of glaciers. It is my concern that readers may think that equations 17/18/19 would constitute a functional model of glacial erosion. This would only be true if there are in fact a lot of physics hidden in Ai - but the need for this is not made clear when the equations are introduced. I urge you to remove this statement entirely. You could, for example, take a path similar to the one that we took. This was simply to point out that models of channel width for glaciers are poorly constrained, and it anyways makes more physical sense that the channel width is proportional to ice flux rather than contributing area - this is because for glaciers there is no connection between contributing area and ice flux due to melting. Therefore, we simply propose equation 13 as an alternative to equation 12 and move forward with that. There is no need to invoke the fairly limiting assumption that there is a constant rate of ice production over the entire upstream catchment.

Sorry, but up to this point the slope of the ice surface has not been replaced by the channel slope of the bedrock. And Eq. (19) is indeed meant to be a functional relationship for glacial erosion, provided that $A_i$ is properly considered as the ice flux converted to its catchment-size equivalent and $S$ is the slope of the ice surface.

Line 93: This is a cool result to be sure! If we apply the sliding only approximation to our equations ($\gamma = 1$) then we recover the same values for the exponents as you have here, which is also cool, though we always consider $S$ to be ice surface slope. We found that this approximation lead to fairly substantial misfit between the steady state morphology of a glacio-fluvial profile when compared to the model presented by Prasicek 2020, which has fewer approximations than either topography based model of glacial erosion (yours or ours). However, we use an approximation for mixed sliding and deformation that does not increase the mathematical complexity of the model, yet leads to almost no misfit at steady state between the more complete model of Prasicek 2020 and our topography based model of glacial erosion. We strongly encourage you to use it as it increases accuracy without increasing the difficulty of solving the model. One caveat: we have carefully tested the accuracy of the various approximations at steady-state, but cannot attest to how well the accuracy holds during transient conditions. Very likely the approximate, topography based model exhibits some misfit during transience, but we would still expect the mixed sliding/deformation approx ($\gamma = \frac{2}{3}$) to perform better than either sliding only or deformation only approximation.

The result with the identical exponent is not really surprising since our models are indeed almost the same up to this point except for the treatment of the deformation component. This is why I was a bit disappointed that you were faster with your approach, which is clearly better for analytical considerations. Nevertheless, I like the result that the concavity index is the same as for rivers, and it turns out to be a big advantage in combination with fluvial erosion. Therefore, I still prefer to start from the sliding-dominated regime and bring in deformation at the end as a correction term. **I pointed out this more clearly now (lines 141–151).**

Line 97: Would be nice to have a citation supporting the statement about psi.

**I removed this statement (line 148)** because I only know that some deviations from $\psi = 3$ were found, but I am not sure how relevant these are for alpine glaciers.

Line 99: Can you expand on what exactly this implies for the factor of proportionality in 14? I'm not sure I followed that very well.

The factor of proportionality is what is typically called erodibility. Erodibilities for different $\theta$ (to be more precise, for different $m$ and $n$) cannot be compared. **The text now directly refers to erodibilities (line 150).**

Line 100: I agree, of course that this is the main difference between 2 and 14, but I don't agree with the reason. As you yourself have mentioned, there is a significant advantage to a topographic based model of erosion, and this is likely the real reason that 2 is written in terms of A and not Q. I think that this is an important distinction, because it had to be demonstrated for fluvial landscapes that $Q \ A^p$, where p is close to 1. This jump will, unfortunately, not be so easy for glaciers.

I am quite sure that $A$ was historically used because it could be measured from maps. In principle, it would be much more convenient if the erosion formula was written in terms of discharge, where the erodibility would not contain precipitation implicitly.

Equation 17: I understand the logic behind this equation, and obviously it's mathematically sound. However, I strongly feel that using a variable termed $A_i$ is confusing and maybe a little misleading. It hides the complexity of the relationship between landscape and climate in a glacier network. A parameter $p_0$ can of course be defined, but what is the physical significance of it? In our work, we also had the same urge as you have here - to show the similarity between topography-based models of glacial and fluvial erosion. In the end we chose to define qi = IA, which is analogous to the fluvial equation q = PA. The difference between this approach and 16/17 is that I is undeniably a function of position, where P can be considered a function of position, but can also be approximated as a constant - an approximation that has been tested and shown to be not terrible. The other difference is that A is A with no distinction between fluvial or glacial and therefore $qi\ Ix^{(}1/eta)$. However, this means that the closest one can come to the SPIM is $E = K_g((I(x)A(x))^\theta S_{icesurface}(x))^\ell$. I would strongly encourage you to not use Ai, because it is confusing. Since Ai is effectively unknowable given the setup here, it is also not really a topography-based erosion model at this point anyways. I think you should either stop at qi, and point out that the SPIM can and is often written this way, or adopt an approach similar to the one that we did in Deal and Prasicek 2020.

Equation 19: I think at some point between equation 4 and 19 it needs to be clearly and visibly pointed out that the ice surface slope has been exchanged for the channel slope. The thickness of glaciers is within an order of magnitude of the height of the topography, and therefore the slope of the ice surface and the slope of the channel do not have to match at all - it is even standard for them to have opposite signs as the terminus of the glacier is approached: 1.Alley, R. B., Lawson, D. E., Larson, G. J., Evenson, E. B. & Baker, G. S. Stabilizing feedbacks in glacier-bed erosion. Nature 424, 758760 (2003).

I guess that you find it only confusing because you worked in a group where another terminology is used. For my part, I dislike the version with $q = PA$ or $q_i = IA$ since $P$ and $I$ are mean values over the upstream catchment here. In my opinion, converting discharges and ice fluxes to an equivalent area is much better, and I would also use it in the context of fluvial erosion under variable precipitation. However, I would not urge anyone who prefers the version with the catchment-average precipitation to switch to this terminology. **So I explained the concept in more detail now (lines 153–174),** but prefer not to switch to your terminology.

No, this is only required for the first version of the numerical implementation (Sect. 3), but not in Eq. (19). **It is pointed out now in the beginning of Sect. 3 (lines 187–194).**

Equations 20 and 21: I strongly disagree with the use of the symbols A and Ai here. I do get the equations - the math is solid, it all checks out. There is nothing fundamentally wrong with the math here. However, I feel like these equations result from a little mathematical gymnastics that hinder the intuitive understanding of the model. It seems to me what's being shown here is actually fundamentally q, normalized by a constant po. We could write 20 as $A = \frac{q}{p_0}$, where $q = s * p + \sum q$. The constant po does allow this to be called A, but particularly in the glacial case where po is effectively unknowable a priori, this is not a helpful equality. I think it would be easier to follow and more properly understood if it were referred to as q. One argument to back this up is figure 5: A cannot decrease downstream. That's not how a concentrative network can function. However, Q can and does decrease downstream in some real rivers and most real glaciers. The other argument is that A is a topographic parameter - it should be calculated from the geometry of the landscape with no concern for things such as precipitation rate. The need to include p, pi and po in equations 20 and 21 is a giveaway that they are not really calculating area.

See response above. Your arguments why this notation should be better do not really convince me.

Line 134: 'over an area around to the cardinal flow path' – wording is a bit awkward, can drop the to'.

In fact rather a mistake than an awkward wording. **I fixed it (line 213),** thanks!

Line 136: I think you mean equation 13

Indeed, thanks! **Fixed (line 215).**

Line 137: Change prolonging' to extending'

Ok, I see that the term prolongation, which is some kind of fixed term in some fields of numerics, is not established in this field. **I changed it throughout the manuscript.**

Line 137: This is clearly a place where no standard has been has been proposed or accepted, so it is exciting to consider the possibilities. I feel that you dismiss these two cases perhaps too rapidly. A couple things come up for me – what makes them endmember cases? It's not clear to me on what spectrum? What variable is reaching an end value? Also what makes them unrealistic? It seems to me that with single timesteps of potentially hundreds to thousands of years converging immediately to U-shaped valleys is not necessarily immediate or unrealistic. This would be the analogous case to the SPIM, where the channel shape cannot evolve, but is immediately specified and whose erosion rate is constant across the entire channel. I understand that the situation is different here because channels are no longer subgrid, but still this seems to me one of the more reasonable choices. Two questions come up for me when thinking about how to handle channel width. First, even though channels are no longer subgrid, is there really enough spatial resolution in the channel that modeling nonuniform erosion rates across channels would be at all realistic? Second, is it worth the effort? What effect would nonuniform channel erosion or non U-shaped valleys have on the largescale evolution of the landscape? I can imagine that right as the channel network develops these dynamics could play a role, but after the network topology is mostly established (e.g. just after the first few timesteps – long before steady state) does it really impact ice routing or erosion rate at a scale greater than say a few channel widths?

Line 140: change a prolongation of the' to extending the'

I see that it is not as trivial for the readers as I thought. However, the point you raised goes a bit in a wrong direction. The approach is not about modeling lateral variations in erosion rate realistically. And the question is not whether anything is worth the effort, it is just simple and works well practically. Same erosion rate for all points does not change the across-valley shape, so just produces a more rapid incision of the V-shaped valley. Same elevation would lead to patches of the same elevation, which replace the flow pattern towards the central line by a parallel flow pattern over long times, and this is inconsistent with the line model. Additionally, it will cause problems when sediment transport comes into play since an enormous amount of sediment is brought to the central line within a single time step. So we need something where a U-shaped valley is some kind of equilibrium shape, and where the erosion rate is higher as long as the valley floor is still to steep for the U-shape, so that it approaches the U-shape through time. And apart from this, Harbor et al. (1988, Nature) even suggested an order of magnitude of 100,000 years for the adjustment. The scheme that I propose needs some 10,000 years for the largest glaciers, so even shorter. Nevertheless, the approach is purely technical without a physical basis. So it is nice that the time scale of adjustment is not obviously totally wrong, but this may be coincidence. **I added a more detailed description and discussion of the three versions (lines 217–243) including the new Fig. 1.**

**I changed it throughout the manuscript.**

Line 142: Again, I am guessing the final handling of channel width is not critical for the behaviour of the model – however, I don't agree that extending the ice flux $A_i$ to the channel edges is the most realistic approach. If I have understood correctly, the parameter $A_i$ is the volumetric flux of ice in the entire channel – the idea of extending it to the edge is for me a bit confusing. Technically there is only one $A_i$ for the entire channel width at any give location – no extending needed. However, for me to make sense of it I have to go back to the original definition of the erosion rate – a function of the sliding velocity. Therefore the most realistic would be to extend to sliding velocity across the channel – either say that the sliding velocity is constant across the channel, or that there is a known gradient. You have constructed a relationship between volumetric flux and flow depth that then leads to a sliding velocity that was maybe implicitly meant to be the centerline – though this was never stated. Then perhaps in this world, the most realistic would be to recognize that ice won't really be able to sustain topography on its surface, and the elevation of the ice will be close to flat across the channel – then the model valley could be filled with ice until the value Ai is used up, and the erosion rates calculated from the ice surface slopes across the channel and the flow depth across the channel. However what I think all this messing about with across channel erosion rate is really saying is that we don't have enough ice flow physics in this model to calculate across channel erosion rates. I feel that we have seen that the U-shaped valley is a sort of equilibrium channel form where the erosion rate can be constant across the channel. This is supported I would argue implicitly by the existence of U-shaped valleys in all glacial landscapes around the earth as well as some modeling studies (Leith, K., Moore, J. R., Amann, F. & Loew, S. Subglacial extensional fracture development and implications for Alpine Valley evolution. J Geophys Res Earth Surf 119, 6281 (2014).). Accepting this channel form a priori as an equilibrium form similar to the way we accept the channel width-discharge relationship for the SPIM leads to the conclusion, for me, to acknowledge that modelling the evolution of channel width is beyond the capabilites of this model and that the best approach is to extend the erosion rate of the centerline to the channel edges.

I accept that you would prefer a different concept here, but I hope that the **more detailed description (lines 217–243) including the new Fig. 1** makes it clear to the readers why this idea cannot work. And in my concept, $A_i$ has to be extended to the swath since the full flux is assigned only to the cardinal flow path for a dendritic flow pattern.

Line 157: I feel that the usage of the value of 1 or 2 for most of the key parameters is strange. This plays the role of a nondimensionalization, but is much harder to follow. For example, $K = 1$ and $U = 1$ for $n = 1$, whereas the 'standard' values ($K = 1e-6$, $U = 1e-3$ for $n = 1$) give a $U/K$ ratio of 1e3 rather than 1. Probably this doesn't matter, and the landscape can just be vertically scaled – but then I, as the reader, have to do all that thinking about it. I must try to interpret the model parameters in terms I am familiar with, and the whole time I am wondering if that dramatically different $U/K$ ratio really doesn't matter at all. I feel that this is mental effort more effectively conducted by the author, and I would really appreciate either a proper nondimensionalization, or simply the usage of more familiar parameter values.

Looks as if I forgot the nondimensionalization since it is always the same. **It is described now in lines 264–271.**

Equation 22: what is A here? Is it just $s + \sum_{donors} s$ or is it equation 20? If it is equation 20, why does p come into $A_{i,eff}$?

Equation 22: I think you need to go into a fair bit more detail about how you arrived at this equation. I feel that this is an important point, but I don't understand why A comes into the ice flux. This seems to me like it would cause the ice to not melt fast enough below the ELA because there would be a portion of the ice flux that is always going to increase downstream regardless of elevation. Also I can't understand at all why the valley shape would change as a function of $A_{i,eff}$ – can you provide some insight into that?

Equation 22 is in fact much less important than it looks. The goal is just to reduce the extended ice flux $A_{i,eff}$ for the points not on the cardinal flow line in such a way that parallel flow is avoided. $A$ in Eq. 22 is the catchment-size equivalent of the total flux. In principle, the static catchment size would also work if the precipitation is in the order of magnitude to $p_0$. **It is described now a bit more in detail (lines 272–287).**

Line 174: I think you mean equation 13 again...

Indeed, **fixed (line 289).**

Line 208: Why is $K_g = 2K$? Why is it larger, and why 2 times larger specifically? I feel that this statement implies conclusions that have already been made about glacial erosion being more erosive than rivers. Why not just leave $K_g = 1$ in the absence of any specific knowledge about $K_g$? Otherwise one could argue that it should actually be 3, or 10 or 100... Alternatively, there are empirical estimates of Kg out there.

$K_g = 1$ might indeed be nice because we would not have to interpret too much difference between the fluvial and the glacial topography. $K_g = 0$ also since the glaciated part would just be uplifted then. Alternatively, $K_g = \infty$ would reproduce the glacial buzzsaw quite well. **I added a remark on the choice $K_g = 2K$ (lines 324–325),** although I cannot imagine why anyone should worry about this choice.

Figure 5: It should be specified clearly that x is distance from outlet of river.

**Added in the captions of Figs. 6 and 8.**

Line 224: I think that the statement about the glacial profiles being steeper is fairly disingenious. It is contingent on choosing the right elevation as your reference point. In fact, across the whole profile the glaciers are actually less steep. Also much of the glacial profile is less steep, the only reason that they are steeper for given reference elevations is because of the steep steps in them. As you state yourself, the only reason there is a step is because of the ice dynamics at the terminus – so the second step in those profiles is actually a transient feature! Comparing transient fluvial profiles and concluding something about their erosion efficiency would be considered a fairly flawed analysis. Also to be fair, even the first step is partly a function of having only sliding and partly a function of not considering the ice surface slope. From what we've seen in our models, there is sometimes a step, and often no step – and when the sliding only condition is imposed, there is much more likely to be a step. Also there doesn't need to be a step at all because it is actually the ice surface slope that drives erosion rate, and while it is true that the ice surface slope goes to infinity, this results in a rapid thinning of the glacier, which often manifests in a channel slope with a sign opposite to the that of the ice surface slope – and therefore no step. Finally, while this step may be partly the result of unrealistic model conditions (that ice flux goes to zero) , this can be handled with a harmless solution like saying that the glacier ends when the ice is just a few meters thick, instead of waiting for it to go to zero. Similar to how the linear stream power law implies infinite elevation, but this is clearly not the case.

The argument about the steepness said "steeper than the fluvial part on average almost up to the ELA". Comparing the $\chi$-transformed profiles to the straight line in Fig. 8 (revised version) shows that this argument is true. **Anyway, I adjusted this section and avoided all statements about what appears to be realistic to me (lines 320–407).**

Line 226: Also, while I agree that requiring infinite ice surface slope at the glacial terminus is unrealistic, I am not convinced that the steps are unrealistic. I think this would be a good place to support this statement with empirical evidence. I would argue that steps / steep fronts under glaciers or at glacial termini are actually quite common – and how can we (yet) say where they come from?

See previous comment.

Line 227: I also agree that erosion by meltwater is important – but you hardly need my agreement for that – there is plenty of evidence, outside of the existence/non-existence of steep fronts to support this conclusion. I think it would be good to cite that here.

**Added the review paper by Alley et al. (2019) (line 357).**

Line 231: I like the idea of this, but can you do better than assuming? There is work on subglacial channel erosion, can some of it be cited here to show that it is at least reasonable to claim an equation of the form of 25?

Not so easy as far as I can see. Qualitatively, it is reasonable that a relation should look like this one if we replace the hydraulic gradient by the slope of the ice surface. The model of Beaud et al. (2016), e.g., also uses the same relations for the incision as typically used for fluvial channels. Nevertheless, there is not enough knowledge about the topology and the scaling properties of meltwater channels, and empirical results on the seasonal variations of discharges and sediment fluxes for individual glaciers are not really helpful here. **So I just stated that the fluvial relation is adopted since there is apparently no better simple model available (lines 358–367).**

Equation 26: Seems like there is an i subscript missing here on the first $A$

**Fixed (line 369), thanks!**

Lines 230–245: I like the simplicity of these equations, but I think you need to be more open about the fact that at this point, equation 27 is wildly unconstrained. Can one even begin to put somewhat empirical values to these $K$s? Either that or bring in some more empirical work on subglacial fluvial erosion.

Rather not, which holds for all the $K$s occurring in the model. Obtaining serious estimates that could be valid, say, for typical alpine glaciers, is quite difficult. And the consequence would be that some of the readers would claim that the model is totally wrong because the do not agree to this parameter value, while others would adopt it without any critical consideration. The advantage of this formulation is that all $K$s have the same meaning. So it is immediately clear what, e.g., $K_f = 0.5K$ would imply. However, I neither want to contribute to the discussion how important meltwater erosion is nor to the discussion whether the glacial buzzsaw exists by suggesting any estimates for the $K$s.

Line 245: "Since there is no discontinuity in the erosion rate at the glacier terminus then, the changes in the flow pattern are much smaller than for the version without erosion by meltwater. The respective profiles depicted in Fig. 7 (dashed lines) reveal a smooth transition from the glacial regime to the fluvial regime". It is clear to me from the text that you find this to be preferable to the previous case with the steep front. I understand this intuition, but I feel obligated to ask why, precisely, is this better? There have been no observations brought out to show that this is really more realistic. How well studied are glacial channel profiles really? Can we really fairly say we already have an intuition for what they should look like? Keeping in mind also that the glacial channels we see today are far, far out of steady state due to us being in an interglacial period. I don't really feel that the section on subglacial fluvial erosion has to be removed or anything, but I do think that the validation of this process based on any attributes of the resulting profiles is potentially folly, and should be avoided. I think you should be more empirically motivated when assessing how realistic these profiles are. It would be sufficient for the development of the model to point out that subglacial fluvial erosion clearly happens – and make some citations, and then, while being open about how unconstrained the parameter values are (unless it is possible to constrain them given existing theory and empirical observations), impartially discuss the differences in the resulting profiles. Or perhaps it is possible to show that the value of $K_f$ is relatively unimportant for the dynamics, that would also be nice.

Figure 7/8 – can you confirm that the value of $K_f$ is the same as $K$ somewhere in the text or figure caption?

Line 256: I think in order to make a statement like this, you need some observations to support it.

I think that I would not be able to show that the value of $K_f$ is relatively unimportant for the dynamics. A moderate variation would probably not have a big effect, but I got the impression that there is a huge uncertainty about the relevance of erosion by meltwater. Anyway, since I refuse to provide any estimates of the $K$s, I agree that an impartial comparison of the versions with and without meltwater erosion is the safest solution. **I rewrote this section accordingly and avoided all statements about what appears to be realistic to me (lines 329–407).**

**Done (lines 377–379 and caption of Fig. 9).**

Really? Typically it is assumed that all properties depend on everything in the world, and here we need a reference? **But ok, I mentioned the Egholm et al. (2012) paper where this was discussed (lines 364–367).**

Line 262: "Let us assume that the identical relations for glacial and fluvial erosion do not only hold for the detachment-limited end-member, and that the shared stream-power model provides a reasonable description of both glacial erosion and erosion by meltwater." – This strikes me as a huge assumption. At this point we have not talked about glacial sediment transport at all. I know that for rivers, both models of erosion and sediment transport are based on the fluid shear stress at the bottom of the river, and even without considering the decades that have gone into studying these processes, there are good reasons to think that they would be related to some degree. Is this the case with glaciers? To be honest, I don't really know how glacial sediment transport works, so I can't comment on it – but it would be, I think, important to see some empirically based arguments, even if they are fairly hand-wavey, as to why we might believe that glacial transport would also be proportional to $KA^{m+1}S^n$.

This is, of course, a huge assumption. However, the shared stream-power model does not imply that glacial sediment transport is proportional to $A^{m+1}S^n$ in general. It rather describes how transported sediment reduces the ability to erode, which is qualitatively similar to the approach used in the iSOSIA model. Both approaches are not constrained well by empirical evidence quantitatively. It can, however, be expected that the effect of transported sediment is smaller for glacial erosion than for fluvial erosion. So applying the shared stream-power model to the glacial component is just an option where the effect vanishes in the limit $K_{t.g} \to \infty$. **This is described in more detail in the revised version (lines 410–433).**

Line 321: I'm not sure if this is a reasonable justification. It is akin to saying that if $E = KA^mS^n$ would lead to no steady state when slope goes to zero, so elevation will grow out of control. The answer is not to remove the slope dependence of erosion rate, but to recognize the limits of the model, and that a purely detachment driven model with no explicit fluid dynamics will never find a slope that is zero at steady state.

Sorry, but it is not removing the slope dependence from the erosion rate, but only from the parameterization of the thickness. And the statement is true as it was written there. **Anyway, I adjusted the discussion and explained it in more detail now (lines 488–514).**

Section 6: I think it would be important to see some sort of comparison between the non-slope dependent ice thickness you implement in the implicit version of the model and the real ice thickness implemented in an explicit version of the model to show that the approximation is reasonable and recovers an ok answer most of the time.

I guess that this goes into the direction of your new approach that you mentioned above. However, combining a slope-dependent thickness with the dendritic flow pattern in the swath causes some problems. Beyond this, I am not convinced that the results of such a comparison would be as useful as it may seem. It cannot be a good approximation over the entire range of slopes, but the parameters could be adjusted to get a reasonable approximation in the range that it most interesting, e.g., for the occurrence of overdeepenings. **I extended the explanation of the parameterization (lines 488–514),** but did not implement an explicit scheme just for performing the comparison.

**Reviewer 2 (Marc Jaffrey)**

First, there is no conflict of interest: My work and the author's work are categorically different and furthermore address different spatial and temporal domains. My work is theoretical with the development of an analytical model of glacier erosion rates at the basin scale, while the author's work addresses glacier erosion at smaller spatial scales utilizing a heuristic approach to define a glacial erosion law for numerical implementation, as is standard for numerical approaches.

Next, please let me express my regret for the tone of my earlier comments without elucidating clear points for the author to address. Since the work makes assumptions in direct conflict with current theories of glacier dynamics, I will not address the computational aspects of the work as the premises on which the model is built are problematic and in this reviewer's opinion unrealistic rending the erosion law in equation 19 open to sever question. Restating the main issues:

The author has not cited nor sufficiently discussed existing literature: This is an issue as the proposed scientific contribution this work cannot be understood except but subject matter experts.

The paper indeed starts from a certain level of fluvial landform evolution and glacier dynamics.

The theoretical underpinnings, the glacier dynamics, in several key place are incorrect in section two and three rendering the hypothesized glacier erosion law, the equation 19, the critical key equation of the numerical model, open to question.

One of the most concerning issue in terms are some of the scientific conclusions of the paper. Most notably in 225: "These findings support the idea that erosion by meltwater must play an important role, at least in the lower part of glaciers where the flux of water is much higher than the ice flux." This statement cannot be justified by the a numerical model regardless of soundness of the theoretical underpinnings. While the contributions of numerical simulations to the research and progress of understanding glacier erosion cannot be overstated, drawing such definitive conclusions about mechanisms of glacier erosion is out of reach for this type of approach.

For me "support the idea that" is not is not a "definitive conclusion about mechanisms of glacier erosion". And even it it was, it would be a completely unimportant part. **Anyway, this statement has been removed (lines 349–350).**

The paper is not written to address the glacial community which I presume is a key target audience for the work.

It is rather written for the landform evolution modeling community, and we will see whether parts of the glacial community are open to simplifying models instead of improving them further with regard to the contributing processes.

Based on these issues I cannot recommend publication of the work as in my opinion the prosed erosion law is unjustified calling into question the numerical model and its results.

See response to the specific points below.

Focused Comments for Section 1:

1–5: "In contrast to fluvial erosion, however, glacial erosion has not been extensively considered in modeling large-scale landform evolution"

"Modeling large-scale landform evolution" should refer to numerical 2D models, so neither individual valley cross sections nor longitudinal profiles of individual glaciers. **Anyway, I added some of your suggested references.**

Here is a list of reference the author might consult:

Egholm, D. L., et al. On the importance of higher order ice dynamics for glacial landscape evolution. Geomorphology 141 (2012): 67-80.

Although I like the work of David Egholm very much, I did not cite this paper because it was not fundamentally new compared to the Egholm et al. (2011) paper where the higher-order approach was already presented.

Harbor, J., 1989. Early Discoverers XXXVI: W J McGee on glacial erosion laws and the development of glacial valleys. Journal of Glaciology 35, 419425.

A quite nice paper about the historical perspective. **It is referenced in the introduction now (line 15).**

Harbor, J., Hallet, B., Raymond, C. A numerical model of landform development by glacial erosion. Nature 333, 347349 (1988).

Although not a large-scale landform evolution model in the sense I considered it, **the references are included now (lines 80–81).**

Harbor, J. Numerical modeling of the development of U-shaped valleys by glacial erosion. GSA Bulletin 104, 13641375 (1992).

Herman, F., Beaud, F., Champagnac, J.D., Lemieux, J-M., Sternai, P. Glacial hydrology and erosion patterns: A mechanism for carving glacial valleys. Earth and Planetary Science Letters, 310, 498508 (2011).

Basically an extension of the ICE-CASCADE model, but **the reference is included in the context of glacial hydrology now (lines 87–88).**

MacGregor, K.R., Anderson, R.S., Anderson, S.P., Waddington, E.D. Numerical simulations of glacial-valley longitudinal profile evolution. Geology 28, 10311034 (2000).

Since it is still a 1D model, it does not fit well into the line of the introduction.

Oerlemans, J. Numerical experiments of large-scale glacial erosion. Zeitschrift fuer Gletscherkunde und Glazialgeologie 20, 107126 (1984).

Although not a large-scale landform evolution model in the sense I considered it, **the reference is included now (lines 80–81).**

Tomkin, J.H. Numerically simulating alpine landscapes: The geomorphologic consequences of incorporating glacial erosion in surface process models. Geomorphology 103, 180-188 (2009).

Basically the ICE-CASCADE model.

Ugelvig, S.V., Egholm, D.L., Iverson, N.R. Glacial landscape evolution by subglacial quarrying: A multiscale computational approach. J. Geo. Res. E. Sur. 121, 2042-2068 (2016).

Quite specific for the context of this paper.

Ugelvig, S. V., et al. "Overdeepening development in a glacial landscape evolution model with quarrying." AGU Fall Meeting Abstracts. Vol. 2013. 2013.

Quite specific for the context of this paper and a conference abstract.

35–40: "A comparable representation of glacial erosion where the erosion rate can be directly computed from properties of the topography is not yet available."

Unfortunately, I have no idea what to do with your comment about the spatial scales.

It is essential discuss the scales, both temporal and spatial, on which erosion rates are being considered. Though not explicitly discuss in the introduction, it can be implied from context that the spatial scales considered are smaller than the scale of the glacial landforms which is sub basin scale.

Alley, R. B., K. M. Cuffey, and L. K. Zoet. "Glacial erosion: status and outlook." Annals of Glaciology 60.80 (2019): 1-13.

Alley, R. B., et al. "Stabilizing feedbacks in glacier-bed erosion." Nature 424.6950 (2003): 758-760.

Andrews J.T. Glacier power, mass balances, velocities and erosion potential. Zeitschrift fur Geomorphologie 13, 1-17 (1972).

Boulton, Geoffrey S. "Processes and patterns of glacial erosion." Glacial geomorphology. Springer, Dordrecht, 1982. 41-87.

Boulton, G. S. "Theory of glacial erosion, transport and deposition as a consequence of subglacial sediment deformation." Journal of Glaciology 42.140 (1996): 43-62.

Cook, Simon J., et al. "The empirical basis for modelling glacial erosion rates." Nature communications 11.1 (2020):

Delmas, M., Calvet, M., Gunnell, Y. Variability of quaternary glacial erosion rates a global perspective with special reference to the Eastern Pyrenees. Quat. Sci. Rev. 28, 484498 (2009).

Hall, Adrian M., et al. "Glacial ripping: geomorphological evidence from Sweden for a new process of glacial erosion." Geografiska Annaler: Series A, Physical Geography (2020): 1-21.

Hallet, B. Glacial abrasion and sliding: their dependence on the debris concentration in basal ice. Annals of Glaciology 2, 23-28 (1981).

Hallet, B. Glacial quarrying: A simple theoretical model. Annals of Glaciology 22, 18 (1996).

Iverson, N.R. A theory of glacial quarrying for landscape evolution models. Geology 40, 679682 (2012).

Menzies, J., Jaap JM van der Meer, and W. W. Shilts. "Subglacial processes and sediments." Past glacial environments. Elsevier, 2018. 105-158.

Steinemann, Olivia, et al. "Quantifying glacial erosion on a limestone bed and the relevance for landscape development in the Alps." Earth Surface Processes and Landforms 45.6 (2020): 1401-1417

Ugelvig, S. V., et al. "Glacial erosion driven by variations in meltwater drainage." Journal of Geophysical Research: Earth Surface 123.11 (2018): 2863-2877.

This looks a bit like a shopping list with your preferred literature, and I found that most of these references address rather specific questions with limited relevance for this paper.

Equation 4 is incorrect: See chapter 7, eq's 7.6, 7.10, 7.17, 8.35, 8.36, 8.65 and sections 8.1, 8.4, 8.5 and 8.6 in Cuffey, Kurt M., and William Stanley Bryce Paterson. The physics of glaciers. Academic Press, 2010. Sliding velocity cannot be reduced to ice thickness and slope under any approximation. As Cuffey and Patterson in discussion the Shallow Ice Approximation say in chapter 8, "The rate of basal slip must be specified directly or through a relation to bed stress such as Eq. 8.25"

Equation 4 just requires $v_s \sim \frac{\tau^\psi}{\sigma}$ (Budd et al., J. Glacial., 23, 157-170, 1979) where $\tau$ = shear stress and $\sigma$ = effective normal stress. According to the shallow ice approximation, $\tau \sim hS$ ($h$ = ice thickness, $S$ = slope of the ice surface). In this form, it yields $v_s \sim \frac{h^\psi}{h-p}S^\psi$ where $p$ = water pressure. Among the references you provided, at least Harbor, Hallet & Raymond, Nature 333, 347-349 (1988) and Tomkin, Geomorphology 103, 180-188 (2009) assumed $p \sim h$, which exactly yields Eq. 4, $v_s \sim h^{\psi-1}S^\psi$. You may question the assumption that the fluid pressure at the bed is proportional to the ice thickness and claim that coupling with a distinct model for melting and for the flow of melt water may yield better results, and that such models are available. However, your statement "Sliding velocity cannot be reduced to ice thickness and slope under any approximation" is wrong. **Anyway, it is explained a bit more in detail now (lines 64–71).**

Equation 6: Yes erosion laws of this form are typically implemented, however there are many other forms that have been used within numerical simulation.

Would you request me to discuss all forms that were ever used?

50: This section requires substantially further discussion and justification. Ice thickness is not a diffusion process. See section 8.5.5 and equation 8.65, 8.70, 8.77, 8.78 and 8.79 in Cuffey and Patterson. In the Shallow Ice Approximation ice flux q h not the partial derivative of h wrt to x. There is a divergence relationship, first order partial derivatives, but diffusion is second order in the spatial partial derivatives. Without substantial justification, ice thickness cannot be treated as diffusion with strong diffusivity.

Sorry, but the shallow water equations and its derivates (shallow ice equations and Savage-Hutter equations for granular flow) assume a free surface and hydrostatic pressure in vertical direction. Then the horizontal force is proportional to the gradient of the free surface and not to the gradient of the bed (as you presumably assume in your reasoning). As a consequence, the velocity in the shallow ice equation also depends on h and on the gradient of the ice surface. Then the divergence term from the mass balance yields second-order spatial derivatives of h in total. The resulting spatial differential operator is elliptic, so that the entire time-dependent equation is a parabolic equation (= diffusion type for the broader readership). Anyway, this statement does not affect the following model development at all.

Section 2:

Again equation 4 is unfounded so that section 2 begins with a false premise. This problem then follows through into equations 8, 9, and the key equation 14 rendering it unfounded.

I am afraid that we have completely different opinions here since I still cannot see which aspect should be wrong.

70-80: This section requires further explanation.

Why? It is very similar to the considerations of Prasicek et al. (2021) and Deal & Prasicek (2021).

100: The authors treatment of catchment size, precipitation, and discharge may have a clear rationale for fluvial systems, but it is not clear why this would apply to glacial system except perhaps at the terminus. The author needs a detail justification.

I cannot see why this should not apply to the glacial mass balance. **Anyway, it is described in more detail now (lines 152–174).**

Equation 19: Taken as a whole, the validity of the proposed erosion law is questionable.

I am afraid that we have completely different opinions here since I still cannot see which aspect should be wrong.

Best regards,

Stefan Hergarten

---

## Author Response (AR2)

Dear Wolfgang,

thanks a lot for your support! I am happy to see the paper on a good way.

Eric Deal criticized that some of the limitations should be made much more clear, and ideally be discussed in more depth. Reading the manuscript, I was also a bit surprised to reach section 8, the conclusions, without a discussion. Of course, many points to be discussed are already mentioned throughout the text, but I think that nevertheless, an iteration of these points would complete the otherwise well written manuscript. One important point that you might want to address is whether the explicit treatment of glacial width could also be applied to rivers whose width is larger than the cellsize? In addition, it might be important to stress that the model is largely restricted to valley glaciers but may have difficulties if glaciations involve plateau or piedmont glaciers. Moreover, it may be useful to point out potential use cases of the model. The explicit treatment of valley shape may preclude studies that may wish to investigate form-process feedbacks at the scale of valleys. Rather, I feel that the model and its implementation will particularly have its merits when studying regional-scale effects of glaciations and their interactions with tectonics.

Hm yes – makes sense to me. I am always afraid of repeating previous parts, so I tried to bring in some aspects at were not explicitly discussed before (lines 520–550). I also thought about the application to wide rivers, but I am not sure whether it would work well since wide rivers often occur in regions of deposition. Forcing the entire swath to follow the erosion of the cardinal flow line works well, but the scheme would not distribute the incoming sediment over the swath automatically. Maybe a student can test this, but it seems to be safer not to mention it here.

177: Minor point, yet I was a bit confused by the term 'cardinal flow line', which I first interpreted as the flow line to the four direct neighbors rather than the diagonal neighbors. Please feel free to make this point clearer if you think this could lead to some confusion.

I would not have thought about this, but I agree that readers might start from the four nearest neighbors and then misinterpret the cardinal flow line. I think it would become clear in Fig. 2 anyway, but nevertheless it is a goof idea to clarify it (lines 177–178).

217: have to be modified

Fixed (line 218), thanks!

241: remove one 'not'

Fixed (line 242), thanks!

Fig. 9: It's really difficult to visually detect topographic changes in Fig. 9 compared to Fig. 7. Consider a hillshade overlain by a grid of total erosion.

Fig. 10: Same as for Fig. 9.

Indeed! For Fig. 9, however, it is mainly a stronger erosion at high elevation that shadows the details even if I plot the differences. So I was not successful in improving the information in Fig. 9. In turn, it indeed makes things much better in Figs. 10 and 12, in particular in combination with plotting a smaller part of the domain. It even works well without hillshading because the valleys and ridges are easily recognized. The modified Fig. 10 also required some extension of the text (lines 395–405).

Best,

Stefan

Stefan Hergarten